# Thermally stable threshold selector based on CuAg alloy for energy-efficient memory and neuromorphic computing applications

Xi Zhou [1,2,3,7], Liang Zhao [2,4,7] ✉, Chu Yan[2], Weili Zhen[5], Yinyue Lin[1,3], Le Li[1,3], Guanlin Du[1,3], Linfeng Lu[1,3], Shan-Ting Zhang[1,3,6], Zhichao Lu[4] & Dongdong Li [1,3,6] ✉

As a promising candidate for high-density data storage and neuromorphic computing, cross-point memory arrays provide a platform to overcome the von Neumann bottleneck and accelerate neural network computation. In order to suppress the sneak-path current problem that limits their scalability and read accuracy, a two-terminal selector can be integrated at each cross-point to form the one-selector-one-memristor (1S1R) stack. In this work, we demonstrate a CuAg alloy-based, thermally stable and electroforming-free selector device with tunable threshold voltage and over 7 orders of magnitude ON/OFF ratio. A vertically stacked $64 \times 64$ 1S1R cross-point array is further implemented by integrating the selector with $SiO_2$-based memristors. The 1S1R devices exhibit extremely low leakage currents and proper switching characteristics, which are suitable for both storage class memory and synaptic weight storage. Finally, a selector-based leaky integrate-and-fire neuron is designed and experimentally implemented, which expands the application prospect of CuAg alloy selectors from synapses to neurons.

In the era of artificial intelligence (AI) and carbon neutrality, the demand for energy-efficient computing systems capable of solving data-intensive computing tasks is surging rapidly. For example, state-of-the-art machine-learning models such as Generative Pre-trained Transformer-3[1] or switch transformers[2] can easily incorporate multiple billions of computing parameters. Conventional computing hardware based on von Neumann architecture experiences major difficulty processing such data-centric workloads, primarily due to the bottle-neck of data transfer between the processor and the memory blocks in these systems (also called the "memory wall" problem)[3].

In order to break the memory wall and achieve energy-saving green AI, the design philosophy of compute-in-memory (CIM) has attracted significant interest[4–7]. Such non-vonNeumann computing systems are often realized with emerging memory technologies such as memristors[5,8], phase change memories[9–11], ferroelectric memories[12] or magnetic memories[13]. In particular, CIM chip based on memristors (or resistive randomaccess memory, RRAM) is one of the most widely studied candidates due to its advantages of low-power operation, low-cost manufacturing and compatibility with complementary metal oxide semiconductor (CMOS) technology[14–16]. In order to achieve RRAM-based CIM with high storage capacity, cross-point array is a favorable scenario in terms of unit cell area (~$4F^2$, where $F$ is the minimum feature size)[6,17]. However, the cross-point arrays of 2-terminal memory devices typically suffer from the

[1]The Interdisciplinary Research Center, Shanghai Advanced Research Institute, Chinese Academy of Sciences, 99 Haike Road, Zhangjiang Hi-Tech Park, 201210 Pudong, Shanghai, China. [2]College of Information Science and Electronic Engineering, Zhejiang University, 38 Zheda Road, 310007 Hangzhou, China. [3]School of Microelectronics, University of Chinese Academy of Sciences, 19 Yuquan Road, 100049 Beijing, China. [4]Hefei Reliance Memory Ltd., Bldg. F4-11F, Innovation Industrial Park Phase II, 230088 Hefei, China. [5]High Magnetic Field Laboratory, Chinese Academy of Sciences, 230031 Hefei, China. [6]Zhangjiang Laboratory, 100 Haike Road, Zhangjiang Hi-Tech Park, 201210 Pudong, Shanghai, China. [7]These authors contributed equally: Xi Zhou, Liang Zhao. ✉e-mail: lzhao2020@zju.edu.cn; lidd@sari.ac.cn

sneak-path current problem, which significantly limits the feasible array size[18–20].

The one-selector-one-memristor (1S1R) architecture, as a general scheme for high-density cross-point memory arrays, is able to suppress the sneak-path currents while improving the storage density[6,21–25]. An ideal selector for cross-point arrays features a small leakage current in the OFF state, sufficiently low resistance in the ON state, steep switching slope (SS) as well as tunable threshold voltage ($V_{th}$) that can match the memristors for joint operations[26]. As of today, selector devices based on insulator-metal transition (IMT)[27,28], ovonic threshold switching (OTS)[29,30], Cu-containing mixed-ionic-electronic conduction (MIEC)[31,32] and metal-filament-based threshold switching[33,34] have been considered for 1S1R integration. The IMT selectors with $NbO_x$ or $VO_2$ switching layer do not require electroforming but have relatively high leakage currents and are susceptible to ambient temperature change, making it difficult to achieve large array operations[27,28]. OTS selectors also exhibit limited selectivity (~$10^3$), and their high-temperature stability for backend-of-line (BEOL) integration is yet to be demonstrated[29,30]. MIEC-based selectors possess a high ON/OFF ratio and promising integration potential but exhibit relatively gradual SS[31,32]. Finally, metal-filament selectors have sufficiently small leakage currents and abrupt switching but often lack stability under high-temperature annealing[35]. In particular, Ag-based metal-filament selectors suffer from the self-agglomeration of Ag under BEOL thermal budget[36,37], whereas Cu-based selectors typically require higher electroforming voltages before normal operations[34]. Therefore, new selector technology with high-temperature stability, electroforming-free feature, steep SS and suitable ON and OFF currents is highly desired.

Furthermore, selectors and 1S1R arrays have potential applications in neuromorphic computing, which adopts certain features of the biological neural systems to accelerate processing and mimic the human brain. For example, spiking neural networks (SNN)[16,38] and Hopfield neural networks (HNN)[5,39] based on memristor crossbars have been widely explored. SNN uses pulses to encode input information which mimics the working pattern of the brain, potentially offering better energy efficiency for AI computing tasks[9]. HNN based on memristors has been explored for applications such as associative memory[39], pattern recognition[40] and solution of non-deterministic polynomial-time-hard problems[5]. However, for practical SNN/HNN applications, large cross-point arrays (e.g., 64 × 64 or larger[5,41]) are desired, which share the same sneak-path current problem as cross-point memories, i.e., the initial weight data cannot be properly programmed into large arrays without selectors (See "Methods" and Supplementary Fig. 1 for array simulations)[42]. In this regard, thermally stable selectors with high selectivity (>$10^6$) are necessary but are rarely demonstrated in the form of large 1S1R arrays due to integration challenges[43]. Also, selector devices are solely utilized to implement the synaptic functions so far, while the volatile and hysteresis nature of selector switching is inherently suitable for implementing oscillatory neurons[9,44].

In this work, we demonstrate for the first time that copper-silver (CuAg) alloy as an electrode material of selectors exhibits superior thermal stability (400 °C/1 h) compared to either Ag or Cu electrodes, making it compatible with CMOS BEOL processing. The high ON current, large ON/OFF ratio (>$10^7$), electroforming-free feature and adjustable $V_{th}$ of the proposed $CuAg/SiO_2/CuAg$ selector confirm its feasibility for large 1S1R cross-point arrays. Subsequently, a functional 64 × 64 1S1R cross-point array is experimentally demonstrated by vertically integrating the $CuAg/SiO_2/CuAg$ selector with $Pt/SiO_2/TiN$ RRAM, exhibiting significant suppression of sneak-path currents and enhanced computational accuracy as synapses. Furthermore, we demonstrate that the proposed selector can be turned into a compact leaky integrate-and-fire (LIF) neuron by simply adding one resistor and one capacitor in parallel, which is a rigorous physical analog of the LIF

neuron model. These results suggest that the CuAg alloy-based selector is a promising and reliable new candidate for cross-point memory and neuromorphic computing applications.

## Results and discussion
### CuAg alloy-based selector with high-temperature stability
Figure 1a, b demonstrates the device concept and working principles of the Cu/Ag metal-filament-based selector device. These two metals can be injected into the intermediate dielectric layer and form a conductive path when applying a sufficient electric field, and the switching can be volatile due to agglomeration and surface-tension effects[45,46]. In this study, the CuAg alloy is adopted as the electrode material[34], which is previously known for its tunable optical properties[47], outstanding mechanical strength[48], durability and oxidation resistance[49,50]. Here, the thermal stability of symmetrical Ag/dielectric/Ag, Cu/dielectric/Cu and CuAg/dielectric/CuAg cross-point selector arrays are first investigated comparatively. The Cu, Ag and CuAg are prepared as bottom electrodes (BE) and top electrodes (TE) by magnetron sputtering. The dielectric layers of $SiO_2$ (Fig. 1) prepared by electron beam evaporation and $Al_2O_3$ (Supplementary Fig. 2) prepared by atomic layer deposition (ALD) are both investigated (See "Methods"). Before annealing, $Ag/SiO_2/Ag$, $Ag/Al_2O_3/Ag$, $CuAg/SiO_2/CuAg$, $CuAg/Al_2O_3/CuAg$ and $Cu/Al_2O_3/Cu$ selectors all exhibit steep threshold switching characteristics with various $V_{th}$ (Fig. 1c and Supplementary Fig. 2b). For $Cu/SiO_2/Cu$ though, its selector behavior is not ideal since it has excessively strong retention (Fig. 1c).

In order to simulate the compatibility of the devices with the BEOL processes, the devices are subjected to a high-temperature annealing process (400 °C, Ar atmosphere, 3 mTorr, hold time 1 h), and their root-mean-square roughness ($R_{RMS}$) is determined by atomic force microscopy (AFM). As shown in Fig. 1d and Supplementary Fig. 2a, the Ag electrodes exhibit significant self-agglomeration after annealing. The $R_{RMS}$ of the annealed device increases tremendously compared to the initial $R_{RMS}$ (from 3.46 to 8.58 nm for $Ag/SiO_2/Ag$ and from 5.03 to 8.37 nm for $Ag/Al_2O_3/Ag$). In contrast, the stacks of $Cu/SiO_2/Cu$, $Cu/Al_2O_3/Cu$, $CuAg/SiO_2/CuAg$ and $CuAg/Al_2O_3/CuAg$ maintain similar morphology before and after annealing, in which the $R_{RMS}$ changes from 2.55, 2.61, 2.79 and 2.21 nm to 3.14, 3.13, 3.78 and 3.87 nm, respectively. In addition, the annealing process significantly degrades the threshold switching behaviors of Ag/dielectric/Ag and Cu/dielectric/Cu devices. The Ag-based devices become open due to the apparent degradation of electrodes. On the other hand, the $V_{th}$ of Cu-based devices significantly increases (Fig. 1e and Supplementary Figs. 2c and 3), potentially caused by copper oxidation even though the structure seems intact. Intriguingly, the CuAg-based device still maintains the threshold switching characteristics after annealing. The crystalline structures of the Ag, Cu and CuAg thin films (~200 nm), which are deposited on $Si/SiO_2$ substrates, are subsequently characterized by X-ray diffraction (XRD, Fig. 1f). The combined results of XRD, scanning transmission electron microscope (STEM, Supplementary Fig. 4a) and corresponding energy dispersive X-ray spectroscopy (EDS) (Supplementary Fig. 4b, c) indicate that the CuAg film is an alloy with an interplanar spacing of 2.30 Å and a Cu/Ag atomic ratio of 3:5 (denoted as CuAg(3:5), if not otherwise specified, CuAg(3:5) is expressed as CuAg in simplified form in this work). In addition, we also vary the process conditions to obtain two more copper-silver alloys, with the Cu/Ag atomic ratios determined as 8:3 and 4:7, respectively (denoted as CuAg(8:3) and CuAg(4:7), see Supplementary Fig. 5a–c for details). After 400 °C annealing, these devices with alloy electrodes also maintain the threshold switching behaviors, indicating a wide process window for good thermal stability (Supplementary Fig. 5d–f).

Furthermore, we explore the leakage current, voltage tunability and endurance of the CuAg-based selectors. Symmetric $CuAg/SiO_2/CuAg$ cross-point architecture is prepared as illustrated in Fig. 2a and Supplementary Fig. 6a, d. The $R_{RMS}$ of the $SiO_2$ interlayer (90 nm

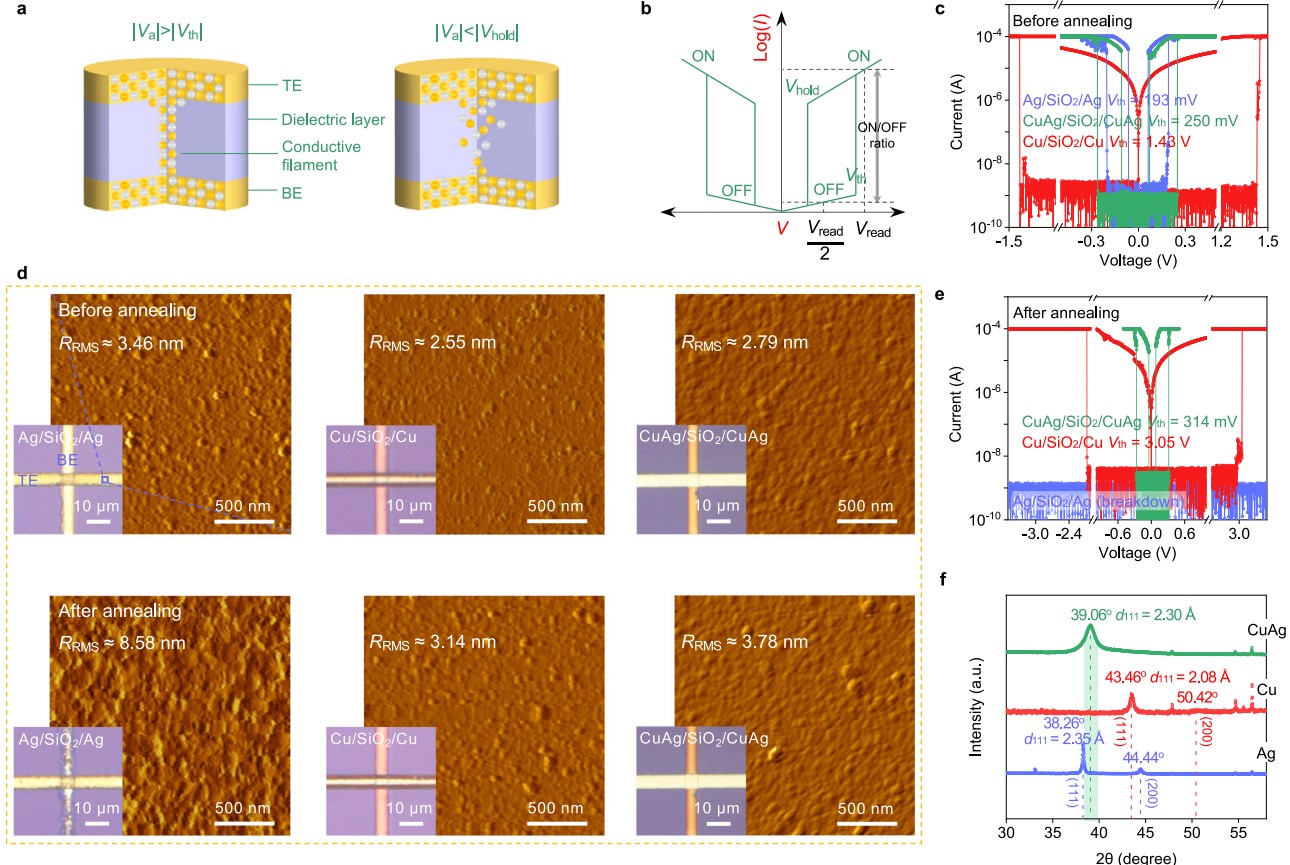

**Fig. 1 | Exploration of the CuAg alloy-based selector. a** Schematic illustrations of a metal-filament-based selector under different applied voltage ($V_a$). **b** Representative current–voltage ($I$–$V$) characteristics of a Cu/Ag metal-filament-based selector, the ON/OFF ratio corresponds to the current variation at the read voltage ($V_{read}$) and half-read voltage ($V_{read}/2$). **c** $I$–$V$ characteristics of Ag/SiO$_2$/Ag, Cu/SiO$_2$/Cu and CuAg/SiO$_2$/CuAg selectors before annealing. **d** Surface morphologies of Ag/SiO$_2$/Ag, Cu/SiO$_2$/Cu and CuAg/SiO$_2$/CuAg devices before and after annealing in Ar atmosphere at 400 °C for 1 h. **e** $I$–$V$ characteristics of annealed Ag/SiO$_2$/Ag, Cu/SiO$_2$/Cu and CuAg/SiO$_2$/CuAg selectors. **f** XRD patterns of the Ag, Cu and CuAg films on SiO$_2$/Si substrates.

thickness) is ~1.95 nm, and the valence state of Si is determined to be dominant Si$^{4+}$ (103.3 eV) (Supplementary Fig. 6b, c). The tunable $V_{th}$ is achieved by varying the thickness of SiO$_2$, where the thicknesses are determined by AFM on patterned SiO$_2$ films (Supplementary Figs. 7 and 8). In order to better evaluate the ON/OFF ratio, a Keithley 6430 source meter with higher precision is used to measure the switching characteristics of the CuAg/SiO$_2$(90 nm)/CuAg selector. The CuAg/SiO$_2$(90 nm)/CuAg device demonstrates stable symmetric threshold switching characteristics with a superior SS of <0.3 mV decade$^{-1}$ and an average $V_{th}$ of 316 mV (Standard deviation σ ≈ 55 mV) when the compliance current ($I_{CC}$) is set to 100 μA (Fig. 2a and Supplementary Fig. 8d). The device's leakage current is at least smaller than 10$^{-11}$ A, the ON/OFF ratio is larger than 10$^7$ (Fig. 2a), which enables large crosspoint arrays that are very difficult to achieve with other categories of selector technologies. Also, the endurance of CuAg selectors can reach over 10$^{10}$ (Supplementary Fig. 9). Moreover, it should be pointed out that the as-fabricated CuAg/SiO$_2$/CuAg selectors do not require an electroforming process with a voltage higher than $V_{th}$. This phenomenon can be explained by the lower migration barrier of Ag/Ag$^+$ in SiO$_2$ compared to Al$_2$O$_3$, which is calculated by ab initio simulations with the nudged elastic band method[51] (Supplementary Fig. 10). In addition, the EDS mappings (Supplementary Fig. 6d) show the diffused Cu and Ag particles, corroborating that the threshold switching of CuAg alloy-based selectors originates from metallic conductive filaments[15,34].

As mentioned above, high-performance selectors need to have sufficiently low leakage current in the OFF state and high drive current in the ON state so as to suppress sneak-path currents and achieve high-

density arrays on the one hand, and to allow easy memory write and read operations without significant voltage drops on the selector on the other hand. In these regards, the CuAg/SiO$_2$(90 nm)/CuAg selector is potentially a promising candidate due to its negligible leakage current (<10 pA), high ON current (>100 μA), steep SS (<0.3 mV decade$^{-1}$), sufficient endurance (>10$^{10}$), electroforming-free feature and superior thermal stability.

## 64 × 64 1S1R array for synaptic weight storage

To further explore the feasibility of applying CuAg/SiO$_2$/CuAg selectors in 1S1R arrays, we construct vertically stacked 64 × 64 1S1R array with CuAg/SiO$_2$/CuAg selectors and Pt/SiO$_2$/TiN memristors. Independent Pt/SiO$_2$(40 nm)/TiN memristors are also prepared and measured for comparison (See "Methods"). TiN electrode is obtained by reactive magnetron sputtering with N$_2$/Ar flow of 0.2/20 for optimized electrical conductivity (Supplementary Fig. 11). The device characteristics of Pt/SiO$_2$/TiN memristor are summarized in Fig. 2b, which exhibit typical bipolar resistive switching behaviors with moderate SET and RESET voltages[52–54].

For 1S1R integration, the manufacturing processes are shown in Supplementary Fig. 12. The as-fabricated CuAg/SiO$_2$/CuAg/TiN/SiO$_2$/Pt 1S1R device exhibits the desired DC sweep characteristics (Fig. 2c). The CuAg/SiO$_2$/CuAg selector acts as a threshold switch with low leakage (<10$^{-11}$ A), significantly suppressing the sneak-path currents in the cross-point array. As the sweep voltage (from CuAg TE to Pt BE) increases, the current of the 1S1R device first sharply increases at ~0.3 V ($V_{th}$), completing the threshold switch (arrow 1, Fig. 2c). Then, a second

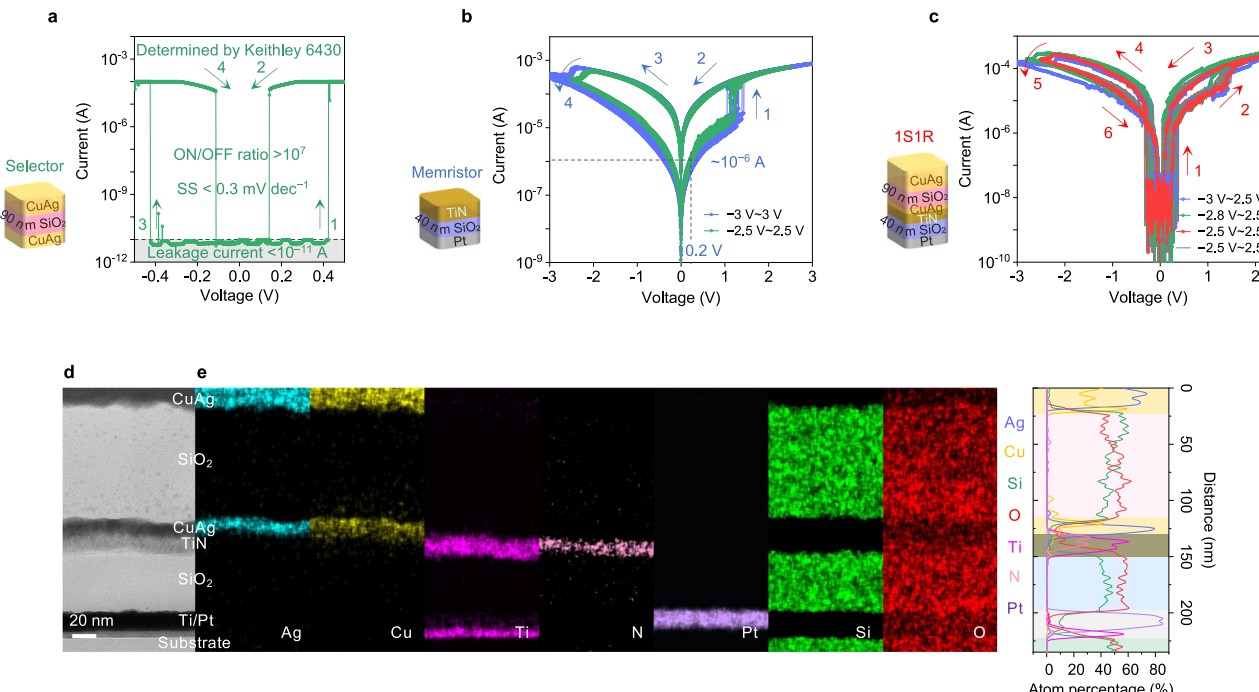

**Fig. 2 | SiO₂-based selector, memristor and 1S1R device. a** *I–V* characteristics of the CuAg/SiO₂(90 nm)/CuAg selector as determined by Keithley 6430. **b** *I–V* characteristics of the Pt/SiO₂/TiN memristor in DC voltage sweep cycles with different stop voltages, the Pt/SiO₂/TiN memristor exhibits typical SET and RESET processes. **c** *I–V* characteristics of the 1S1R device in DC voltage sweep cycles with different stop voltages. **d** Cross-sectional STEM image of one 1S1R device. **e** The EDS mapping and linear sweep results with the elements Ag, Cu, Ti, N, Pt, Si and O corresponding to (**d**).

current jump occurs at -1.5 V, indicating the SET process (arrow 2). When the voltage sweeps back to -0.1 V (hold voltage, $V_{hold}$), the current drops, and the selector device switches to the OFF state (arrow 3). When the voltage sweeps to negative values, the selector turns on again at ~−0.3 V (−$V_{th}$, arrow 4), followed by a reduction in current at above −2 V, indicating the RESET process (arrow 5). Finally, after the negative voltage sweeps back to ~−0.1 V (−$V_{hold}$), the current drops again, and the device returns to the HRS (arrow 6). A full switching cycle is hence completed. To visualize the stacking of 1S1R devices, the cross-sectional profile is extracted by focused ion beam milling in the middle of one 1S1R device, where the stacking order of the electrodes and dielectric layers can be clearly observed using STEM and EDS (Fig. 2d, e).

In addition, the potential of the 1S1R array for implementing next-generation memory and neuromorphic computing primitives is considered. The structure schematic, chip and array photos are shown in Fig. 3a–c. By optimizing the RRAM interlayer process, various devices in the as-fabricated 64 × 64 1S1R array exhibit expected electrical properties, making it a promising candidate for high-density memory applications (Fig. 3d and Supplementary Fig. 13). With the integration of the selector, the leakage current of the 64 × 64 1S1R reduces from 10⁻⁶ (64 × 64 1R in Fig. 3e) to <10⁻¹¹ A (Supplementary Fig. 14). The sneak-path current and parasitic capacitance are significantly suppressed (Supplementary Fig. 15), indicating that 1S1R array is a particularly useful technology for SNN applications with improved operation speed and reduced power consumption. In summary, the ON/OFF ratio of 1S1R devices achieves an improvement of 10⁵ times relative to the Pt/SiO₂/TiN 64 × 64 1R array alone, reducing the power consumption and improving the feasible array size as cross-point memory.

Furthermore, we demonstrate the advantages of applying 1S1R to synaptic weight storage by performing simulations of vector matrix multiplication (VMM) using 32 × 32 and 64 × 64 cross-point arrays, with and without selectors (See "Methods" for array simulations). Figure 3f shows a schematic of the simulation procedure, in which the input

vector and binary weight matrices are randomly generated[55–58]. The weights are encoded in the form of RRAM conductance matrix (*S*) in which LRS corresponds to '1' and HRS corresponds to '0'. During the simulations, the LRS resistances are generated using the measured distribution, and the ON/OFF ratios of RRAM and 1S1R are assumed to be 100 and 10⁷, respectively. The output results in terms of BL currents are simulated with one fully connected (FC) layer of 64 × 64 or 32 × 32 weight matrices, as shown in Fig. 3g and Supplementary Fig. 16. These results indicate that the arrays with selectors are able to generate much more similar output feature maps to the theoretical values than those without selectors. In order to quantify the accuracy of VMM computation, the correlation coefficient of the simulated output vector (IR drop and sneak-path currents considered) versus the theoretical output (by floating-point calculation) is calculated. The probability density of the correlation coefficients obtained from 1000 sets of random inputs are shown in Fig. 3h. It can be concluded that cross-point arrays with selectors achieve much higher VMM accuracy compared to those without selectors (93.8% vs. 48.05% for 64 × 64 array). Subsequently, the accuracy of VMM calculations using the 1S1R and 1 R subarrays is also compared in Supplementary Fig. 17 to demonstrate the positive effect of the 1S1R on the VMM. It can be seen that the accuracy decreases significantly with increasing array size in the absence of the selector. By eliminating the sneak-path currents, the as-fabricated 1S1R device can strongly suppress the accuracy degradation and enable much larger arrays of synaptic data to be accessed simultaneously, boosting energy efficiency.

## Selector-based LIF neuron

The LIF neuron is an important classical biological neuron model which has been widely studied and adopted to mimic the human brain[59] (Fig. 4a). The LIF model features a "leaky" resistor and a capacitor connecting in parallel with a switch, the voltage across which represents the membrane potential of the biological neuron (Fig. 4b). So far, there have been many attempts to emulate LIF model with

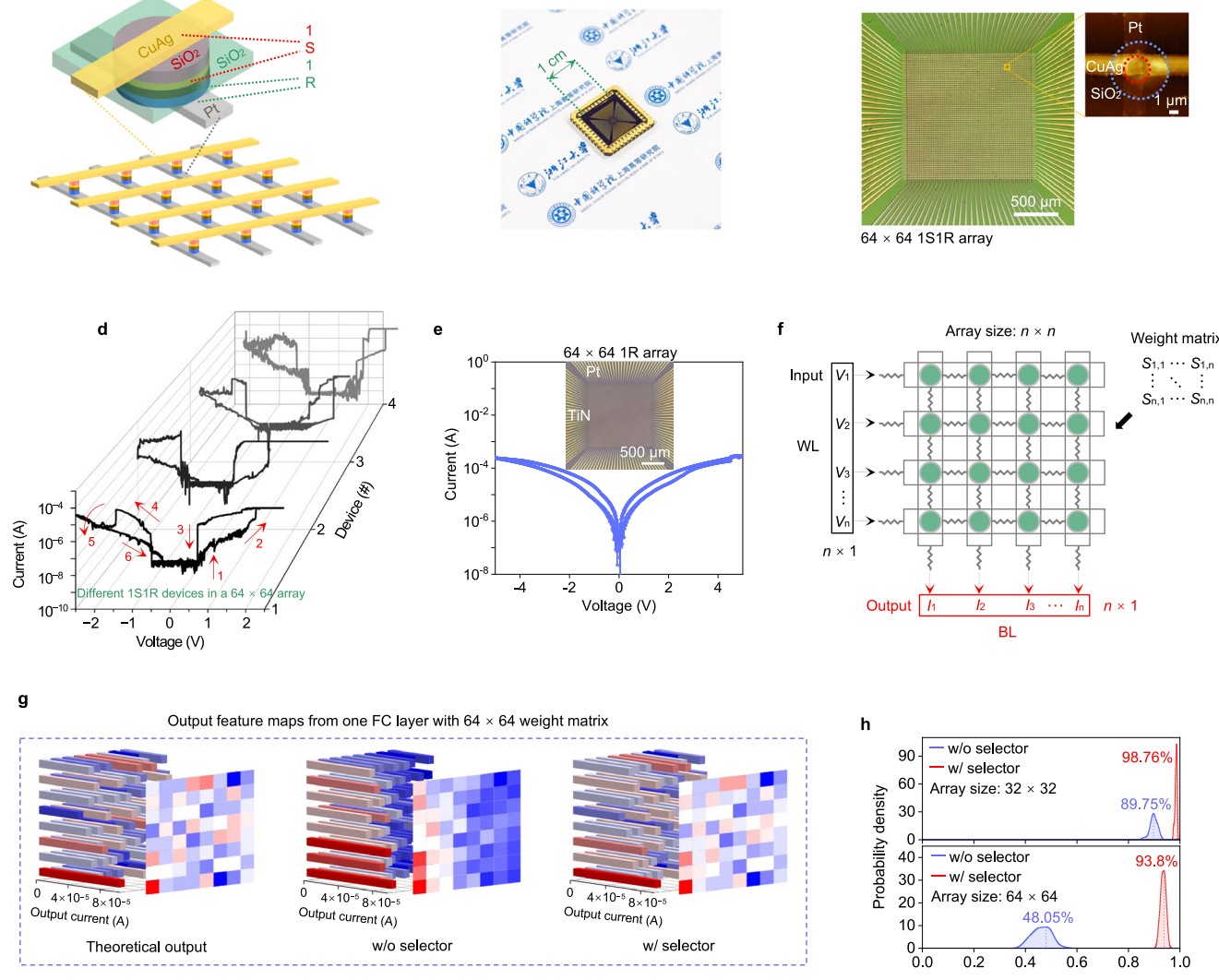

**Fig. 3 | The 64 × 64 1S1R array. a** Schematic illustration of the integrated 1S1R devices. **b** Photo of the integrated 64 × 64 1S1R chip. **c** Optical micrograph of the 64 × 64 1S1R array. The inset shows an AFM image of one 1S1R device. **d** *I–V* characteristics of different 1S1R devices in a 64 × 64 array. **e** *I–V* characteristics of the Pt/SiO₂/TiN memristor measured from the 64 × 64 1R array. The inset shows the optical image of the 64 × 64 1R array. **f** Schematic diagram of VMM simulation using cross-point array, where the voltage vector input to the word line (WL) is a random value and the current after VMM is output from the bit line (BL). **g** Output feature map obtained by VMM simulation using one FC layer with 64 × 64 weight matrix for the theoretical output (left), without selector (middle), and with selector (right), respectively. **h** Probability density of the correlation coefficients between the theoretical results and the output results obtained by generating 1000 random sets of voltage vectors and weight matrices fed into the RRAM matrix with and without selector.

CMOS analog circuits[60] or non-volatile memories such as NOR Flash[61] or FeFET[62]. Figure 4c depicts a key feature of the LIF neuron: there is minimal input for the neuron to reach the threshold and fire, and once the threshold is reached, the firing frequency increases almost linearly with increasing input. By setting the refractory period ($\tau_0$), RC time constant ($\tau_{RC}$) and threshold current ($I_{th}$), the variation curve of firing rate with input current in Fig. 4c is simulated. Compared to other selectors with higher leakage (e.g., VO₂[28], NbOₓ[10], or OTS-based[30]), the extremely low leakage currents of CuAg alloy-based selectors is the key enabler for implementing a LIF neuron. This is because the equivalent leaky resistance of the LIF neuron circuits depends on both the parallel resistor and the OFF state resistance of the switch. With the connection topology of Fig. 4b, the CuAg alloy-based selector's OFF state resistance and its impact on the parameters of the LIF neuron is negligible compared to the parallel resistor, where the value of the parallel resistor can be well controlled in modern integrated circuit design.

In order to characterize the behaviors of the proposed LIF neuron, we carry out electrical measurements using the setup shown in Fig. 4d

and Supplementary Fig. 18. When a constant current is input to the neuron, it will charge up the capacitor and increase the input node voltage from 0 V, which in turn will induce leakage current through the parallel resistor. If the input current is smaller than the $I_{th}$ (Fig. 4e), the input node voltage will saturate at a voltage smaller than the $V_{th}$ of the selector, and the neuron will not be fired. On the other hand, if the input current is above $I_{th}$, the input node voltage will rise above $V_{th}$, leading to an ON state of the selector device (i.e., the neuron is fired). The firing of the neuron manifests itself as a high transient current across the device and the discharge of the parallel capacitor. Based on the mechanism described above, the $I_{th}$ of the LIF neuron can be derived in terms of the selector $V_{th}$: $I_{th} = V_{th}/R$. The firing frequency of the selector-based LIF neuron also increases with the input current due to the less time needed to charge up the parallel capacitor again. In summary, the LIF behaviors predicted by the theoretical model have been experimentally observed from the selector-based LIF neuron. We may conclude that the proposed LIF neuron circuit based on CuAg alloy selectors is a near-perfect physical analog of the LIF model.

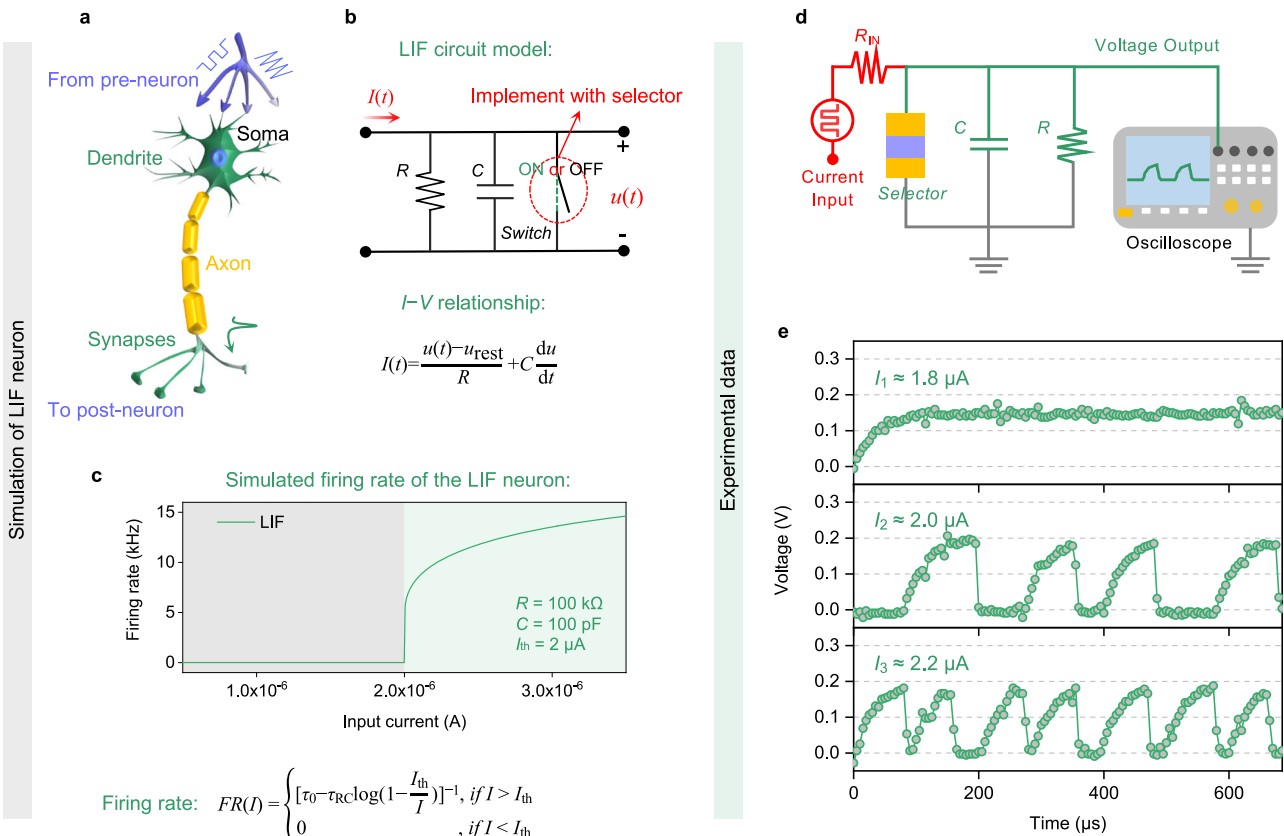

**Fig. 4 | On the validation of selector-based LIF neuron. a** Schematic diagram of a biological neuron. **b** Circuit model of a LIF neuron, the relationship between the $I(t)$ and the $u(t)$ is described by the equation in (**b**), where $u_{rest}$ is the resting membrane potential. **c** Simulation schematic of a LIF response function, where the relationship between the firing rate and the input current is described by the equation in (**c**). **d** Circuit for implementing the selector-based LIF neuron. **e** Experimental demonstration of the LIF neuron's firing rate with various input currents ($I_1 \approx 1.8\ \mu A$, $I_2 \approx 2.0\ \mu A$ and $I_3 \approx 2.2\ \mu A$).

In summary, we have demonstrated the CuAg alloy-based selector as a promising candidate for high-density cross-point memory and neuromorphic computing applications, which features simple preparation processes, good thermal stability, electroforming-free selector behaviors, tunable $V_{th}$ and over 7 orders of magnitude ON/OFF ratio. Based on this selector device, the proper 1S1R device characteristics in a vertically stacked 64 × 64 1S1R cross-point array are achieved, including sufficiently low sneak-path current, desirable $I–V$ curves, stable memory window and switching endurance. Such cross-point arrays can be used to store the synaptic weights of neural networks and achieve more accurate and energy-efficient in-memory computation for AI. A selector-based LIF neuron is also experimentally demonstrated, providing a new perspective for the application of such devices as neurons. The CuAg-alloy electrode selector has good thermal stability that is compatible with the CMOS BEOL process. It can potentially realize the on-chip integration of 1S1R array and LIF neuron, which implements two different functions (synapse and neuron) on one technology platform.

## Methods
### Device fabrication

(1)   CuAg-based selector: The Cu, Ag and CuAg BE are deposited on polished $SiO_2$ (300 nm) on Si wafers by means of standard photolithography and magnetron sputtering (AJA, ACT Orion 8). Cu and Ag are obtained by magnetron sputtering of 50.8 mm diameter Cu target (99.99% purity) and Ag target (99.99% purity), respectively. During the co-sputtering process, the Ag target is sputtered at a radio frequency power of 60 W/120 W/240 W, and the Cu target is sputtered at a direct current power of 60 W. $SiO_2$

films with different thicknesses are obtained by electron beam evaporation with an acceleration voltage of 10 kV at room temperature (99.99% purity of $SiO_2$ particles, evaporation rate: ~5 Å/s). The $Al_2O_3$ layer is deposited on the BE by the ALD process (200 °C, 60 cycles). For a single cycle of ALD, trimethylaluminum (TMA) is first pulsed to 70 Pa for 0.02 s, followed by a 15 s purge. $H_2O$ is then pulsed to 90 Pa for 0.01 s, followed by a 20 s purge. After that, Cu, Ag and CuAg alloy thin films as TE are deposited by photolithography and magnetron sputtering.

(2)   Annealing process: All selectors are placed in an argon atmosphere (3 mTorr) at a heating rate of 0.3 °C per second to 400 °C and maintained for 1 h, followed by slow cooling to room temperature.

(3)   $Pt/SiO_2/TiN$ memristors: Patterned Ti/Pt (5/15 nm) as BE are deposited at room temperature by means of photolithography and electron beam evaporation (99.99% purity of Ti and Pt particles). $SiO_2$ films are obtained by electron beam evaporation with an acceleration voltage of 10 kV at room temperature (99.99% purity of $SiO_2$ particles, evaporation rate: ~5 Å/s). Patterned TiN as TE is deposited at room temperature by sputtering (AJA, ACT Orion 8) Ti target (99.99% purity) in $N_2/Ar$ flow ratio of 0.2 sccm /20 sccm (3 mTorr) at room temperature.

(4)   The 1S1R array: The 64 × 64 1S1R array consists of $CuAg/SiO_2/CuAg$ selectors and $Pt/SiO_2/TiN$ memristors stacked vertically, and the fabrication steps are detailed in Supplementary Fig. 12.

### Materials characterizations
Optical microscope images are obtained by 3D laser scanning confocal microscope (Keyence VK9710K). AFM images and Raman spectra are obtained by a combined AFM/Raman (532 nm) instrument (NT-MDT

NTEGRA). The composition and structural analyses are carried out by XRD (Rigaku D/max2200) and X-ray photoelectron spectroscopy (XPS, Thermo Fisher 250 XI). STEM images and corresponding EDS are obtained by FEI Titan Themis 200.

### Electrical measurements

Electrical characterizations are executed with an Agilent B1500A semiconductor device parameter analyzer, a Keithley 6430 source meter, an Agilent MSO7054A oscilloscope, a Keysight 33250A waveform generator, a Keithley 4200 SCS, a Keithley 707A switch matrix, and a self-made variable resistance box (10 k, 100 k, 1 M and 10 MΩ).

### Array simulations

The input parameters of the $n \times n$ cross-point array simulations include voltage vector applied to the WL $[V_1, V_2, V_3,..., V_n]$, the weight data in the form of conductance matrix corresponding to all cross-points $[S_{1,1},..., S_{n,n}]$, and the line resistances between two adjacent junctions along WLs or BLs ($R_{WL}$, $R_{BL}$). The output parameters of VMM are defined as the current vector read from the BLs $[I_1, I_2, I_3,..., I_n]$ when the BL voltages are fixed at zero. The junction conductance is defined by the measured results of 1S1R devices, and the line resistance is defined with empirical values. The cross-point array simulations are performed as SPICE-style simulations of the equivalent circuits implemented in MATLAB. The steady-state electrical characteristics of a cross-point array can be completely described by the WL plane voltages $[V_{WL}(i,j)]$ and BL plane voltages $[V_{BL}(i,j)]$ at each cross-point, where $1 \le i, j \le n$. Based on Kirchhoff's law and the input parameters, these $2 \times n \times n$ voltage variables can be written in matrix form and solved for the currents in an iterative manner. The accuracy of the VMM operations using the cross-point array is characterized by the statistical results of the correlation coefficient between the simulated output current vector and the theoretical output vector, obtained using multiple sets of randomly generated input parameters (See Supplementary Fig. 1 and Supplementary Table 1 for further details).

## Data availability

The data that support the findings of this study are available from the corresponding author upon reasonable request.

## Code availability

The simulation codes supporting the findings of this study are available from the corresponding authors upon reasonable request.

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

## Acknowledgements

This work was supported by the Natural Science Foundation of Shanghai (No. 19ZR1479100) and the Shanxi Science and Technology Department (20201101012). This work was supported in part by the Key Research and Development Program of Zhejiang Province under Grant 2021C01039.

## Author contributions

L.Z. conceived and designed the experiments and performed the simulation. X.Z. performed the experiments and measurements. L.Z. and X.Z. wrote this paper. C.Y. contributed to the simulation. W.Z. contributed to the experiments and measurements. Y.L., L.Li and G.D. assisted the device fabrications under the supervision of D.L., L.Lu, S.Z. and Z.L. All authors discussed and reviewed the manuscript. D.L. and L.Z. supervised the project.

## Competing interests

The authors declare no competing interests.
