## [Peer Review File · Nature Communications]

REVIEWER COMMENTS

Reviewer #1 (Remarks to the Author):

This manuscript by Zhou et al. reports a selector device based on CuAg alloyed electrode, which is claimed to be thermally stable and electroforming free. The selector device was subsequently used to construct a 64*64 1S1R array and implement a LIF neuron. The strong part of the study, in my view, is the fabrication of a 1S1R array in experiment, but the authors did not use this array for the claimed computing applications, with the VMM operations and fully-connected networks investigated by simulations. Besides, the manuscript has a number of ambiguities and weaknesses, which makes it overall a preliminary study. My technical questions are detailed below:

(1) The work is not organized in a logically coherent study. Since the authors have adopted SiO₂ as the functional layer in the selector, RRAM as well as 1S1R cells, it seems quite strange why Al₂O₃ based selector was built in the first place? Why not directly using SiO₂ for the investigation of electrode material and annealing effect in Fig. 1? Furthermore, the LIF neuron part is quite separated from the rest of the work, and the neuronal functions are lack of novelty compared with many existing works in the literature.

(2) This work has claimed thermally stable and electroforming free characteristics of the alloyed selector, but for selectors endurance is in important consideration, because the selector needs to be turned on every time even for reading operations and regardless of the memory state. Therefore, the requirement on endurance for a selector is higher than that for memory devices. The authors only showed 1S1R operation in 120 cycles (Fig. 3e), but this is not enough. I suggest to characterize the endurance of the selector device. In addition, transient characteristics including turn-on speed and turn-off delay of the selector should also be characterized to warrant fast and reliable operations.

(3) It is difficult to understand why the CuAg/SiO₂/CuAg selector with a large SiO₂ thickness of 90nm does not require electroforming process, and this is also against prior results. Even as claimed in the work the migration barrier of Ag/Ag⁺ in SiO₂ is low, there is no Ag inside SiO₂ with very large thickness. Therefore, an initialization process should be needed to incorporate Ag into SiO₂ first. The rationality behind the electroforming free feature should be better justified.

(4) There exist very large variations in the threshold voltages of the selector, as shown in Fig. S6. How would this affect the 1S1R operation?

(5) It seems the on-state resistance of selector is higher than the on-state resistance of the RRAM cell, which is not the ideal case. As a consequence, the on/off ratio of 1S1R structure is decreased significantly, compared with that of 1R, as shown in Fig. 2. From Fig. 3e one can see that the effective on/off ratio is less than 2 in 1S1R operations.

(6) There seem to be lots of particles in the cross section of the CuAg/SiO₂/CuAg structure in Fig. 2d. What is the reason for that?

(7) Although a 1S1R array was fabricated in experiment, the authors did not use this array for the claimed computing applications. Instead, both the VMM operations and the fully-connected network were evaluated by simulations. These computing functions only involve single-layer operations. I don't understand why they were not performed experimentally. Is there a yield issue in the array? At least experimental results from a sub-array should be provided.

Reviewer #2 (Remarks to the Author):

The authors demonstrated a CuAg alloy-based selector, thermally-stable and electroforming-free selector device with over 7 orders of magnitude ON/OFF ratio. Further, the CuAg alloy-based selector applied in the 64 × 64 1S1R cross-point array and the LIF neuron was investigated. The results above reveal the great application prospect of CuAg alloy-based selectors from synapses to neurons. However, some problems need to be worked out before the publication of the article.

1. In Fig. 1e, the selector devices with Ag and Cu electrodes were damaged and possessed a threshold voltage over 20 V after annealing, respectively. By contrast, the device in which the electrodes featured an Ag/Cu atomic ratio of ~5:3 was demonstrated proper threshold switching characteristics after annealing. How did such atomic ratio determinate the maintenance of the characteristics? If the composition of copper atoms changes from 0 to 1, what happens to the threshold switching characteristics?

2. In Fig. 3d, the electrical properties of a few devices were shown in a 64 × 64 cross-point array. What was the yield of the entire array? In addition, the performance comparisons with and without selectors should be studied in the 64 × 64 1S1R cross-point array experimentally, as a validation of the simulation results. Furthermore, a more interesting application could be demonstrated based on the large-scale array.

Reviewer #3 (Remarks to the Author):

General Feedback:

This paper aims to demonstrate a new application for CuAg/Al₂O₃/CuAg selectors (which have been previously demonstrated in paper ref. 34), as a candidate for building LIF switches for SNN applications.

The paper first highlights how at high temperatures these devices have superior thermal stability to either Ag or Cu selectors with the same dielectric. The authors then continue to discuss the integration of these devices into a 1S1R array with 64x64 elements. Finally, the authors conduct measurements showing how these devices can be utilized as an LIF device, which can be used to construct SNN's.

The Supplemental Figures are also extremely interesting as they cover a number of interesting points such as the optimization of the dielectric thickness and DFT studies highlighting the potential physical mechanism for Ag migration in Al₂O₃.

As a follow-up paper, this work is very interesting. The authors have greatly expanded my understanding of their devices with this paper.

However, this paper is being submitted as an application paper, in which there are gaps in the analysis of the suitability of this array for the intended application.

Given that the title and abstract stress the utility of these devices in certain use cases, I think the paper is incomplete without a discussion of the endurance of this selector. For both memory and SNN applications, the selector endurance is of tantamount importance. Furthermore, for memory applications, a discussion (of citation) of the incorporated RRAM retention and RRAM endurance for the chosen metal combination is also important. (and of course a confirmation would be even better).

Also important is confirmation of the failure mechanism, do these particular selectors fail as shorts or as opens? For memory applications, the open-circuit failure mechanism translates to lost data,

whereas the short-circuit translates to increased sneak paths (which is slightly more tolerable). Depending on the overall design of the SNN, the LIF neuron will either become non-selective (always transmitting input pulses when shorted), or always non-responsive, both failure modes are presumably bad for inference accuracy.

As far as suitability is concerned, the authors should state their endurance targets for these two applications, based on their reading of the literature. For instance, even though examples of OxRRAM have been demonstrated with 10^{10} endurance, there is still concern that this is not sufficient for SNN's and some additional architectural support is needed (see for example <https://ieeexplore.ieee.org/document/9380671>). The selector endurance needs to be compared to the total number of spikes expected in this application, so this reviewer would not be surprised to hear requirements for endurance targets of 10^{12} or even higher.

Without this additional information to demonstrate suitability, I would recommend that this paper be rewritten and resubmitted as an expanded exploration of the device physics.

Furthermore, given the authors are now in possession of large arrays of these devices it should be possible to report detailed endurance and measurements with good statistics.

Detailed feedback:

Line 45: (Nitpicky grammar point). In this particular sense of the word "interest", it is an uncountable noun, and so the pluralization here is grammatically incorrect. It should be just "interest" (not "interests"). Also to my ear the use of "tremendous" is somehow awkward, I would recommend "has attracted significant interest".

Line 53: As far as this reviewer is concerned, all cross-point arrays suffer from sneak-paths. This problem is not unique to ReRAM.

Lines 127-130: It seems that the authors are using single-device statistics to demonstrate the failings of the Ag-only and Cu-only films. This reviewer's experience with R&D clean rooms generate a lot of variability, and one failed device is not enough to judge whether the same recipe should always fail. Some reference of the number of devices tested would build confidence in this result, or barring that, if these effects are expected from the literature for Cu/Dielectric/Cu or Ag/Dielectric/Ag devices, then it would be sufficient to state that these results are in line with previous results.

Line 137: The fact that the AgCu ratio is determined by XRD or EDS to be 5:3, implies that this value is not well controlled (or at least not actively constrained). Is this what one would expect from the sputtering process used here? Is there any data or modelling to prefer any different alloy composition?

Line 159: Reporting the threshold voltages to three significant figures implies strong repeatability and consistency across all tested devices. Looking at the supplemental materials, these thresholds are very stochastic. Please include error ranges (+/- values) with these stochastic quantities. This should be done for all stochastic quantities or quantities which are averaged over many devices.

Lines 165-167: Comment: The level of modelling shown here and in the Supplemental materials is an unexpected bonus in an application paper such as this. Even if this paper were to be reframed as a study of the devices themselves, this analysis would be a very nice addition. This is really nice work.

Side question: Is there any explanation as to why the Cu-only electrodes have low thresholds before annealing, but only require electro-forming after annealing? Given the amount of detail invested in the DFT modelling of the Ag migration, this would be something interesting to include if there is any better understanding of the change taking place in the Cu devices. Are there perhaps some new parasitic interface layers being formed in this case?

Lines 174-176: No mention is made of the importance of endurance or retention, which implies that the quantities listed are all that matters.

Lines 224-245: Some phenomenological explanation would be helpful in understanding the VMM case. For example, is there some “back-of-the-envelope” scaling relationship which relates the 1) cell-resistance, 2) the stray resistances, and 3) the array size to predict the magnitude of the errors here? Such a supplement (or a citation to one) would be nice to have in helping the audience understand these simulations. This may be very complicated, but any intuitive explanation you can provide would be helpful.

Other questions on these VMM sims:

- What do the correlation diagrams look like in the smaller 32x32 case? One can see in the supplemental figure that the the sneak-path currents are not as critical, but how large are they really?

- Are the stimulus voltages applied to be inputs assumed to be uni-polar?

Line 307: A quick document search indicates that this is the only instance of the word “endurance” in the entire body of the text. The authors claim without evidence that the selector endurance is acceptable, without any reporting of the device endurance, and no arguments or citations as to what level of endurance is required for these applications.

Reviewer #1:

This manuscript by Zhou et al. reports a selector device based on CuAg alloyed electrode, which is claimed to be thermally stable and electroforming free. The selector device was subsequently used to construct a 64*64 1S1R array and implement a LIF neuron. The strong part of the study, in my view, is the fabrication of a 1S1R array in experiment, but the authors did not use this array for the claimed computing applications, with the VMM operations and fully-connected networks investigated by simulations. Besides, the manuscript has a number of ambiguities and weaknesses, which makes it overall a preliminary study. My technical questions are detailed below:

(1) The work is not organized in a logically coherent study. Since the authors have adopted SiO₂ as the functional layer in the selector, RRAM as well as 1S1R cells, it seems quite strange why Al₂O₃ based selector was built in the first place? Why not directly using SiO₂ for the investigation of electrode material and annealing effect in Fig. 1? Furthermore, the LIF neuron part is quite separated from the rest of the work, and the neuronal functions are lack of novelty compared with many existing works in the literature.

Response: We thank the reviewer for the valuable comment and have made extensive revisions to improve the logical coherence and clarify the innovations.

As for the reason that Al₂O₃-based selector was built in the first place, it reflected the chronological order of our exploration process: we first discovered that CuAg/Al₂O₃/CuAg selectors exhibited threshold switching and superb thermal stability at high temperatures. Later, we further investigated CuAg/SiO₂/CuAg selectors and obtained similar threshold switching without the electroforming process. Finally, we constructed the 1S1R arrays using CuAg/SiO₂/CuAg selectors and Pt/SiO₂/TiN memristors. Indeed, organizing the results in this order may become less coherent to the readers. Thus, we have revised the manuscript to add the experimental data of CuAg/SiO₂/CuAg selectors before and after 400 °C annealing to improve the logical coherence. As shown in Fig. R1 (Fig. 1c–e in the revised manuscript), similar to the CuAg/Al₂O₃/CuAg selector, the CuAg/SiO₂/CuAg selector maintains stable morphology and electrical properties after 400 °C annealing. At the same time, both Ag/SiO₂/Ag and Ag/Al₂O₃/Ag selectors degrade significantly after annealing. And both Cu/SiO₂/Cu and Cu/Al₂O₃/Cu selectors demonstrate significant threshold voltage increase after annealing. The results of metal/Al₂O₃/metal selectors before and after annealing had been moved to the Supplementary

information (Fig. S2) to improve the logic coherence of the article. The corresponding discussions were also modified to accommodate the reorganized figures (page 5, paragraph 1 in the revised manuscript).

As for the design of LIF neuron, the novelty of using CuAg-alloy electrode selector is that the leakage current is extremely low and the RC time constant of the designed LIF circuits can be more accurately controlled (because compared to other selectors such as OTS or VO₂-based, the parasitic leakage current of our selector can be neglected compared to that of the *R*). Moreover, the CuAg-alloy electrode selector has good thermal stability that is compatible with CMOS BEOL process. Thus, it can potentially realize the on-chip integration of 1S1R array and LIF neuron, and can realize two different functions (synapse and neuron) on one technology platform. Both of these are innovative features of the technology developed in this paper, which are rare in prior art, thus we emphasize these innovations at page 15, paragraph 1 in the revised manuscript.

Fig. R1 (Fig. 1c–e in the revised manuscript). Exploration of the CuAg alloy-based selector. a, Surface morphologies of Ag/SiO₂/Ag, Cu/SiO₂/Cu and CuAg/SiO₂/CuAg devices before and after annealing in Ar atmosphere at 400 °C for 1 hour. **b**, *I-V* characteristics of Ag/SiO₂/Ag, Cu/SiO₂/Cu and CuAg/SiO₂/CuAg selectors before annealing. **c**, *I-V* characteristics of annealed Ag/SiO₂/Ag, Cu/SiO₂/Cu and CuAg/SiO₂/CuAg selectors.

(2) This work has claimed thermally stable and electroforming free characteristics of the alloyed selector, but for selectors endurance is in important consideration, because the selector needs to be

turned on every time even for reading operations and regardless of the memory state. Therefore, the requirement on endurance for a selector is higher than that for memory devices. The authors only showed 1S1R operation in 120 cycles (Fig. 3e), but this is not enough. I suggest to characterize the endurance of the selector device. In addition, transient characteristics including turn-on speed and turn-off delay of the selector should also be characterized to warrant fast and reliable operations.

Response: We thank the reviewer for the valuable comment. We have carried out further experiments to characterize the endurance and switching transient characteristics of the selectors, as shown in Fig. R2 (also Fig. S9 in the revised supplementary information). The CuAg/SiO₂/CuAg selector shows good endurance under AC and DC voltage stimulations, capable of maintaining threshold switching after >10¹⁰ AC pulses or more than 120 seconds of DC stimulation. Limited by the test setup, the device's rise time, fall time and turn-on/turn-off delay times are measured as 2.021, 1.323, 0.722 and 0.379 microseconds, respectively. Consider the RC delays associated with top and bottom electrodes and the measurement system, we expect the actual switching delay time to be much faster than the obtained results, if the test structure can be optimized (e.g., GSG structures). Ultrafast measurements of selector devices will be a topic for our future studies.

Fig. R2 (Fig. S9 in the revised supplementary information). Endurance and temporal behaviors of the CuAg/SiO₂/CuAg selector. a, Pulse endurance and **b,** DC retention of the CuAg/SiO₂/CuAg selector. **c,** Schematic diagram of the rise, fall and turn-on/turn-off delay times measurement setup. The waveform input from the waveform generator (Keysight 33250A) and the waveform output from

the measurement setup are measured by an oscilloscope (Agilent MSO7054A). **d**, Rise, fall and turn-on/turn-off delay times of the CuAg/SiO₂/CuAg selector. The relevant parameters of the devices are only conservatively evaluated due to the circuit structure and test accuracy.

(3) It is difficult to understand why the CuAg/SiO₂/CuAg selector with a large SiO₂ thickness of 90 nm does not require electroforming process, and this is also against prior results. Even as claimed in the work the migration barrier of Ag/Ag⁺ in SiO₂ is low, there is no Ag inside SiO₂ with very large thickness. Therefore, an initialization process should be needed to incorporate Ag into SiO₂ first. The rationality behind the electroforming free feature should be better justified.

Response: We thank the reviewer for the valuable comment. The cross-sectional TEM/EDS (Fig. R3 and Fig. S6d in the revised supplementary information) results show that there are actually many metallic spherical clusters in the low-density SiO₂ layer, the diameters of which are typically between 1 nm and 5 nm. These clusters may contribute to the partitioning of a thick SiO₂ layer into thin regions with a low migration barrier for metal ion transfer among the clusters, thus the devices exhibit electroforming-free behaviors. We speculate that the cause for this type of behaviors is due to the electron beam evaporation process used to deposit SiO₂, which may result in a more porous film. This combined with the low migration barriers of Ag and Cu in SiO₂ (Ag: 0.61 eV; Cu: 0.4~1.1 eV) may contribute to the forming-free selector even with thick SiO₂. If the SiO₂ layer can be prepared using ALD method, it is expected to significantly reduce the required thickness, which will be a topic for future investigations. The corresponding discussion is added in the caption of Supplementary Fig. S6.

Fig. R3 (Fig. S6d in the revised supplementary information). Cross-sectional STEM image of the as-fabricated CuAg/SiO₂/CuAg selector and corresponding EDS elemental mapping of Ag, Cu,

Si, and O. The green circles mark the diffused metallic spherical particles with diameters of 1 to 5 nm.

(4) There exist very large variations in the threshold voltages of the selector, as shown in Fig. S6 (now Fig. S8 in the revised supplementary information). How would this affect the 1S1R operation?

Response: We thank the reviewer for the valuable comment. For cross-point array operation, the most popular scheme is called the V/2 scheme, in which the cells can be divided into three categories: selected, half-selected and unselected. In order to make sure that none of the unselected cells are activated, we want $1/3V_{\text{read}} < V_{\text{th}} < V_{\text{read}}$ (under the V/2 bias scheme) to ensure that half-selected cells are not turned-on. Also, V_{read} should be significantly smaller than V_{set} and V_{reset} (e.g., V_{read} smaller than 1/2 of V_{set} & V_{reset}) to ensure that the cell is not disturbed by read.

For the devices presented in this work, we can see that the above requirements can be easily met: for $V_{\text{th}} = 0.316$ V and $\sigma \approx 55$ mV (Fig. S8d) and RRAM write voltage of 1.5~2.5 V (Fig. S13), we can set V_{read} at 0.6~0.7 V range to meet the requirements of both read and write operations, even with the presence of V_{th} variations. In other words, although the variation of the V_{th} is relatively large, as long as it is still significantly larger than $1/3V_{\text{read}}$ and significantly smaller than V_{set} & V_{reset} , the 1S1R array can still be operated properly. Detailed analysis of the 1S1R operation schemes and variation effects can be found in some prior arts, e.g., S. Kim et al., *IEEE Trans. Electron Devices*, 61, 2820 (2014); P. Y. Chen et al., *IEEE Trans. Electron Devices*, 62, 12 (2015); and B. Song et al., *Appl. Phys. A* 123, 356 (2017). In particular, B. Song et al. discussed in their *Appl. Phys. A* paper the V_{th} variation effects on 1S1R operations in details. They have demonstrated that minimum V_{th} is closely related to OFF- and ON-resistance of selectors while maximum V_{th} is related to ON resistance of selectors. Higher OFF resistance and lower ON resistance will assure that array can stand larger variation of V_{th} without increasing the complexity and power of the operation. Since the selector reported in this work already demonstrated very high OFF resistance, we may further reduce the ON resistance to improve the immunity of 1S1R operation to the V_{th} variations.

Moreover, based on our proposed selector stack, there is also room to further reduce the V_{th} variation by optimizing the process flow. Currently, the effective area of the fabricated selector is $\sim 25 \mu\text{m}^2$, and the bottom electrodes could get bumpy due to the lift-off process. This means that multiple paths which favors filament formation can be formed, and they vary significantly from device to device. This variation makes the location and shape of the conductive filament more random when the selector is

turned on, which leads to the non-uniformity of the V_{th} . If the filament formation can be controlled at fixed paths with a smaller device feature size and planarization processes, it will help reduce the variation of the V_{th} .

(5) It seems the on-state resistance of selector is higher than the on-state resistance of the RRAM cell, which is not the ideal case. As a consequence, the on/off ratio of 1S1R structure is decreased significantly, compared with that of 1R, as shown in Fig. 2. From Fig. 3e one can see that the effective on/off ratio is less than 2 in 1S1R operations.

Response: We thank the reviewer for the valuable comment. Since the CuAg-alloy selector is the focus of this work, the RRAM devices are not fully optimized to match the selector in the early stage.

In this revised work, we found that the ON resistance of the selector can be reduced by increasing the atomic ratio of Ag (see Fig. R4a, before annealing). Together with the optimized RRAM interlayer process in 1S1R, we achieved a larger ON/OFF ratio ($>10^3$, see Fig. R4b) than in the initial submission. With optimized ON current of the selectors as well as better-matched RRAM devices, the 1S1R devices meet the requirements well for cross-point array applications. Thus the $I-V$ characteristics of a series of 1S1R devices in Fig. 3d were replaced by the new data and corresponding description was modified in page 10, paragraph 1 in the revised manuscript.

Fig. R4. a, $I-V$ characteristics of three CuAg alloy-based selectors, where more Ag content delivers a smaller ON state resistance. **b**, $I-V$ characteristics of the optimized 1S1R device in the 64×64 1S1R array.

(6) There seem to be lots of particles in the cross section of the CuAg/SiO₂/CuAg structure in Fig. 2d. What is the reason for that?

Response: We thank the reviewer for the comment. The results of cross-sectional TEM and EDS (Fig. R3 and Fig. S6 in Supplementary information) show that the particles in the SiO₂ layer are metallic Ag and Cu. According to our theoretical calculations and results in the reference (D. M. Guzman et al., *J. Appl. Phys.* 117, 195702 (2015)), Ag and Cu have low migration barriers (Ag: 0.61 eV, Cu: 0.4~1.1 eV) in SiO₂. And the low-density SiO₂ prepared by electron beam evaporation further facilitates the easy formation of spherical metal particles, which in turn reduces the potential barrier for the first switching of the selector. By improving the SiO₂ process in the selector and achieving denser SiO₂, it is possible to reduce the particle caused by metal diffusion while reducing the required SiO₂ thickness.

(7) Although a 1S1R array was fabricated in experiment, the authors did not use this array for the claimed computing applications. Instead, both the VMM operations and the fully-connected network were evaluated by simulations. These computing functions only involve single-layer operations. I don't understand why they were not performed experimentally. Is there a yield issue in the array? At least experimental results from a sub-array should be provided.

Response: We thank the reviewer for the valuable comment. Due to the conditions of our experimental facility which is a university fab, our 64 × 64 1S1R arrays usually do not exhibit perfect yield. In particular, since low-density SiO₂ layers are prepared using electron beam evaporation, it is difficult to control the flatness of the interface after multiple stacks, which is one of the main reasons for low yields. Ideas for improving array yields include reducing thickness and increasing uniformity with ALD SiO₂ as mentioned earlier, using advanced lithography to achieve smaller device linewidth to improve yield and uniformity, and using a Cu/Ag pillar and chemical mechanical polishing process to enhance the flatness of the bottom electrode to enhance uniformity. In the new 1S1R array, we find a 4 × 4 region with 100% yield, as can be seen from Fig. R5 (Fig. S13 in Supplementary information), the 1S1R array has good uniformity. These results further demonstrate the feasibility of the 1S1R array for VMM and neural network acceleration. We understand there are great challenges in accomplishing a fully functional system-on-chip especially in our university fab, but we believe the proposed 1S1R array with improved thermal stability has large application prospects if the process is migrated to commercial foundries with state-of-the-art manufacturing capability.

Fig. R5 (Fig. S13 in the revised supplementary information). I - V characteristics of a 4×4 subarray in a 64×64 1S1R array. By summarizing the results of all experiments, the key to influence the yield of 1S1R array is the control of the bottom and middle layer flatness.

Reviewer #2:

The authors demonstrated a CuAg alloy-based selector, thermally-stable and electroforming-free selector device with over 7 orders of magnitude ON/OFF ratio. Further, the CuAg alloy-based selector applied in the 64×64 1S1R cross-point array and the LIF neuron was investigated. The results above reveal the great application prospect of CuAg alloy-based selectors from synapses to neurons. However, some problems need to be worked out before the publication of the article.

(1) In Fig. 1e, the selector devices with Ag and Cu electrodes were damaged and possessed a threshold

voltage over 20 V after annealing, respectively. By contrast, the device in which the electrodes featured an Ag/Cu atomic ratio of $\sim 5:3$ was demonstrated proper threshold switching characteristics after annealing. How did such atomic ratio determinate the maintenance of the characteristics? If the composition of copper atoms changes from 0 to 1, what happens to the threshold switching characteristics?

Response: We thank the reviewer for the valuable comment. The effect of Cu/Ag atomic ratio on the thermal stability of threshold switching is indeed an important topic worth further investigation. In our first attempt to build an alloy-based selector, we set the alloy ratio by co-sputtering power (Ag target: RF 120 W, Cu target: DC 60 W) and achieved stable threshold switching. As suggested by the reviewer, we carry out additional experiments on two different Cu/Ag ratios, i.e. 8:3 (Cu target: DC 60 W, Ag target: RF 60 W, named as CuAg(8:3)) and 4:7 (Cu target: DC 60 W, Ag target: RF 240 W, named as CuAg(4:7)), where the crystal structures and composition of the alloys are determined by XRD (Fig. R6a), GIXRD (Fig. R6b) and EDS (Fig. R6c).

Subsequently, 5 sets of metal/SiO₂/metal (Ag, Cu, CuAg(8:3), CuAg(3:5) and CuAg(4:7)) selectors were annealed, where the results before and after annealing of the newly added 2 sets of CuAg(8:3) and CuAg(4:7) alloy-based selectors are shown in Fig. R6d–f, and the results before and after annealing of the other 3 sets of the selectors are shown in Fig. R1. After annealing, the introduction of Cu ensures the thermal stability of the CuAg alloy electrodes. For the selectors with the same dielectric layer (90 nm low density SiO₂), the V_{th} and ON state resistance of the devices show the increasing trend with the increase of Cu content. Based on the statistical results of devices, the electrical characteristics of the devices with more Ag content (CuAg(3:5) (Fig. 1 in the revised manuscript) and CuAg(4:7) (Fig. R6e,f)) are stable after annealing with small V_{th} variation. The sample with less Ag content (CuAg(8:3)) also maintains the threshold characteristics after annealing, but the ON state resistance increases significantly, presumably due to the formation of suboxide by Cu drawing oxygen from the dielectric layer.

Fig. R6 (Fig. S5 in the revised supplementary information). Crystal structures of Cu, Ag, and three copper-silver alloy films determined by **a**, XRD and **b**, GIXRD. **c**, The compositions of the two additional CuAg alloy films determined by EDS. **d**, Surface morphologies of CuAg(8:3)/SiO₂/CuAg(8:3) and CuAg(4:7)/SiO₂/CuAg(4:7) devices before and after annealing in Ar atmosphere at 400 °C for 1 hour. **e**, I - V characteristics of CuAg(8:3)/SiO₂/CuAg(8:3) and CuAg(4:7)/SiO₂/CuAg(4:7) devices before annealing. **f**, I - V characteristics of annealed CuAg(8:3)/SiO₂/CuAg(8:3) and CuAg(4:7)/SiO₂/CuAg(4:7) devices.

Combining the results of all devices, the effect of Cu/Ag ratio on thermal stability can be summarized as follows: pure Ag is very unstable due to its soft texture and agglomeration phenomenon, which is not compatible with CMOS processes. The Cu doping in Ag can stabilize the threshold switching phenomenon over a wide range of doping concentration, i.e. threshold switching can be maintained after annealing at 400 °C. However, when the ratio of Cu is high (e.g., CuAg(8:3)), the devices exhibit a potential oxidation phenomenon that causes an increase in the ON state resistance. Such a phenomenon and the large V_{th} increase after annealing of the pure Cu selectors (see Fig. R1b,c) can be explained by the Cu oxidation. Also, the post-annealing changes of ON state resistance and V_{th} at higher Cu concentrations are not fully controllable. This suggests that electrode materials with higher Ag content are preferred (e.g. Cu/Ag between 4:7 and 3:5). The above analysis provides guidance for the selection of the Cu/Ag ratio in CuAg alloy-based selectors. We added relevant descriptions in page 6, paragraph 1, associated with the representative data in the revised supplementary information (Fig. S5).

(2) In Fig. 3d, the electrical properties of a few devices were shown in a 64×64 cross-point array. What was the yield of the entire array? In addition, the performance comparisons with and without selectors should be studied in the 64×64 1S1R cross-point array experimentally, as a validation of the simulation results. Furthermore, a more interesting application could be demonstrated based on the large-scale array.

Response: We thank the reviewer for the valuable comment. Achieving high yields for 64×64 1S1R array is challenge for us in a university fab. Here, the potential of suppressing the memristor sneak path current via selector is demonstrated by electrical behaviors of multiple devices, thus promising high yield large-scale arrays is achievable if the stacking hierarchy is migrated to chip fabricators. As shown in Fig. R5, we have obtained 4×4 subsection with good yield in 64×64 array, indicating the yield is at an acceptable level (the operating voltage in the array is not uniform enough due to manufacturing process limitations, and some devices require very high operating voltages), otherwise the leakage current in an array of this size would be very high. For example, as shown in Fig. R7 (Fig. 3e in the revised manuscript), we have made 64×64 1R arrays, in which the devices have difficulty forming a large operating window due to the high leakage. Our current ideas to improve the yield of 1S1R arrays are: 1. using advanced lithography to achieve smaller effective region, 2. using chemical

mechanical polishing process to enhance the flatness. We use simulations to illustrate the advantages of 1S1R array in VMM computing, and we will further explore these interesting on-chip applications in the future.

Fig. R7 (Fig. 3e in the revised manuscript). **a**, Optical micrograph of the 64×64 1R array. **b**, I - V characteristics of the Pt/SiO₂/TiN memristor measured from the 64×64 1R array.

Reviewer #3:

General Feedback:

This paper aims to demonstrate a new application for CuAg/Al₂O₃/CuAg selectors (which have been previously demonstrated in paper ref. 34), as a candidate for building LIF switches for SNN applications.

The paper first highlights how at high temperatures these devices have superior thermal stability to either Ag or Cu selectors with the same dielectric. The authors then continue to discuss the integration of these devices into a 1S1R array with 64×64 elements. Finally, the authors conduct measurements showing how these devices can be utilized as an LIF device, which can be used to construct SNN's.

The Supplemental Figures are also extremely interesting as they cover a number of interesting points such as the optimization of the dielectric thickness and DFT studies highlighting the potential physical mechanism for Ag migration in Al₂O₃.

(1) As a follow-up paper, this work is very interesting. The authors have greatly expanded my understanding of their devices with this paper.

However, this paper is being submitted as an application paper, in which there are gaps in the analysis of the suitability of this array for the intended application.

Given that the title and abstract stress the utility of these devices in certain use cases, I think the paper is incomplete without a discussion of the endurance of this selector. For both memory and SNN applications, the selector endurance is of tantamount importance. Furthermore, for memory applications, a discussion (of citation) of the incorporated RRAM retention and RRAM endurance for the chosen metal combination is also important (and of course a confirmation would be even better).

Response: We thank the reviewer for the kind feedback and valuable comments. First of all, although this paper is a follow up of our preliminary progress report published as a poster presentation at Device Research Conference 2021 (Manuscript Ref. 34), the major theme has shifted to the discovery of a thermally-stable, forming-free selector device that is more suitable for CMOS integration. In this regard, we finally identify CuAg/SiO₂/CuAg as the selector device of choice, which is a different stack from the one reported in Ref. 34. Based on this new stack, we further explore its potential applications in cross-point 1S1R array and LIF neuron.

For both memory and SNN applications, we fully agree with the reviewer's suggestion that it is important to evaluate the endurance of selector. We have carried out further studies on the endurance, as shown in Fig. R2 (Fig. S9 in the revised supplementary information). The CuAg/SiO₂/CuAg selector shows good endurance under AC and DC voltage stimulation, capable of maintaining >10¹⁰ switching counts and more than 120 seconds of continuous stimulation.

Also, we acknowledge that RRAM's retention and endurance characteristics are important for the cross-point 1S1R application. The primary goal of this work is to integrate 1S with 1R and demonstrate 1S1R functionality. Previous studies have shown the potential for high endurance/retention of SiO₂-based memristor (H. Jiang et al., *Sci. Rep.* 6, 22216 (2016)). As a result, SiO₂-based RRAM device is adopted for 1S1R so that the same dielectric can be used for both RRAM and selector. The experimental results indicate that the SiO₂ interlayers serve the purpose of studying 1S1R behaviors very well. In our future studies, we would switch to other RRAM stacks to make 1S1R reliability even better. In fact, one of the corresponding authors (Dr. Liang Zhao) has been working on commercial RRAM technologies which can meet the retention and endurance requirements of embedded RRAM IP products (L. Zhao et al., VLSI 2022).

(2) Also important is confirmation of the failure mechanism, do these particular selectors fail as shorts or as opens? For memory applications, the open-circuit failure mechanism translates to lost data,

whereas the short-circuit translates to increased sneak paths (which is slightly more tolerable). Depending on the overall design of the SNN, the LIF neuron will either become non-selective (always transmitting input pulses when shorted), or always non-responsive, both failure modes are presumably bad for inference accuracy.

Response: We thank the reviewer for the valuable comment. We found the major type of failures are device short when the selectors are kept in the conductive state and no longer turned off. But overall, the percentage of such short failures is relatively small, otherwise we would not be able to find working 1S1R devices in the 64×64 array due to the significant sneak-path current.

In addition to short failures, the main challenge for cross-point 1S1R yields is that both 1S and 1R voltages have variation (as discussed in the response to 4th question of reviewer #1), and both need to be matched. So it can be difficult to guarantee that all 1S1R devices will work under uniform operating conditions (V_{read} , V_{set} , V_{reset}). The idea of solving this problem is to use mass production type process flow (e.g., reducing device area, using planarization process, etc.) to obtain higher uniformity and yield, which is the direction of our future efforts.

(3) As far as suitability is concerned, the authors should state their endurance targets for these two applications, based on their reading of the literature. For instance, even though examples of OxRRAM have been demonstrated with 10^{10} endurance, there is still concern that this is not sufficient for SNN's and some additional architectural support is needed (see for example <https://ieeexplore.ieee.org/document/9380671>). The selector endurance needs to be compared to the total number of spikes expected in this application, so this reviewer would not be surprised to hear requirements for endurance targets of 10^{12} or even higher.

Response: We thank the reviewer for the valuable comment. We agree that the endurance required for SNN is quite high. Assuming each selector cell is access 100 times per second on average, and the product lifetime is 10 years, we need at least $100 \times 3600 \times 24 \times 365 \times 10 = 3.1536 \times 10^{10}$ cycles of endurance for the selector. We supplement the selector endurance data as suggested by the reviewer, and the selector is able to withstand over 10^{10} read pulses, which has the potential to be used in both memory and neural network applications. Limited by the experimental setup, we haven't finished endurance test of higher cycle numbers, which is a direction for future exploration.

(4) Without this additional information to demonstrate suitability, I would recommend that this paper be rewritten and resubmitted as an expanded exploration of the device physics.

Furthermore, given the authors are now in possession of large arrays of these devices it should be possible to report detailed endurance and measurements with good statistics.

Response: We thank the reviewer for the valuable comment. We have thoroughly revised the manuscript to include more details regarding both the device physics explorations and the suitability for applications.

Regarding device physics explorations, we supplement our results with additional process splits, device test data, as well as materials characterizations (Figs. R1–R6) to obtain a more thorough understanding of the physics of CuAg alloy selectors.

Regarding the suitability for applications, we further demonstrate that the presented selector devices have good endurance up to 10^{10} times, and good 1S1R yield can be achieved from at least a portion of the 64×64 array (4×4 portion, Fig. R5). On the other hand, without the involvement of the selectors, the 64×64 memristor-only array cannot be properly operated (Fig. R7). This proves the effectiveness of the selector for both memory and neural network applications. For future studies, we have also pointed out plan to a few directions (e.g. tuning Cu/Ag ratio, reducing device area, using planarization process, etc.) for further optimization of the device performances. We hope these improvements make this manuscript acceptable to *Nature Communications*.

Detailed feedback:

Line 45: (Nitpicky grammar point). In this particular sense of the word “interest”, it is an uncountable noun, and so the pluralization here is grammatically incorrect. It should be just “interest” (not “interests”). Also to my ear the use of “tremendous” is somehow awkward, I would recommend “has attracted significant interest”.

Response: We thank the reviewer for the careful review. This sentence has been revised accordingly to: “..., the design philosophy of compute-in-memory (CIM) has attracted significant interest.”

Line 53: As far as this reviewer is concerned, all cross-point arrays suffer from sneak-paths. This problem is not unique to ReRAM.

Response: We thank the reviewer for the careful review. This sentence has been revised accordingly

to: “However, the cross-point arrays of 2-terminal memory devices typically suffer from the sneak-path current problem, ...”

Lines 127-130: It seems that the authors are using single-device statistics to demonstrate the failings of the Ag-only and Cu-only films. This reviewer’s experience with R&D clean rooms generate a lot of variability, and one failed device is not enough to judge whether the same recipe should always fail. Some reference of the number of devices tested would build confidence in this result, or barring that, if these effects are expected from the literature for Cu/Dielectric/Cu or Ag/Dielectric/Ag devices, then it would be sufficient to state that these results are in line with previous results.

Response: We thank the reviewer for the valuable comment. The thermal stability judgments for Cu, Ag, and CuAg alloy-based selectors are based on the results of a few (~10) devices showing consistent changes in electrical and morphological properties after annealing. For example, Fig. R8 shows the images of multiple Ag-based selector devices (with either Al₂O₃ and SiO₂ inter-dielectric) after 400 °C annealing. The annealed device electrodes exhibit prevalent agglomeration under microscope. And the electrical results (devices all open) correspond well with the observed morphology change.

Fig. R8. Surface morphologies and I - V characteristics of **a**, Ag/Al₂O₃/Ag and **b**, Ag/SiO₂/Ag devices after annealing in Ar atmosphere at 400 °C for 1 hour. All devices show open characteristics due to Ag self-agglomeration.

Fig. R9 shows the changes of morphological and electrical properties of multiple Cu/SiO₂/Cu selectors. The Cu electrodes seem unchanged after annealing treatments, but all exhibit a tendency of threshold voltage increase (to above 2 V). In addition, the pure Cu electrodes tend to possess strong retention behavior with non-volatile features.

We speculate that this is related to the reaction of Cu with oxide to produce CuO_x. The interfacial CuO_x layer would not only increase the series resistance and threshold voltage, but also weaken the capillary effect of the electrode upon the metallic filament and potentially increase the retention (P. Zhou et al., 2009 IEEE International Memory Workshop). Hence, the fast migration of metal ions in and out of the dielectric layer is suppressed by this interfacial layer, leading to the observed changes (threshold voltage increase, stronger retention and even conversion to non-volatile switching).

Fig. R9. *I-V* characteristics of four different Cu/SiO₂/Cu devices after annealing treatments. The inset shows the optical images of the corresponding Cu/SiO₂/Cu cross points.

Line 137: The fact that the AgCu ratio is determined by XRD or EDS to be 5:3, implies that this value is not well controlled (or at least not actively constrained). Is this what one would expect from the sputtering process used here? Is there any data or modelling to prefer any different alloy composition?

Response: We thank the reviewer for the valuable comment. In the revised version, we have carried out additional experiments to study two extra Cu/Ag ratios and verify their threshold switching characteristics before and after annealing (see Fig. R6 and R10). The Cu/Ag ratio of the alloy films are controlled by tuning the sputtering powers of Cu and Ag, as can be found in the 1st response to reviewer #2. The new experiments demonstrate a consistent trend: pure Ag is very unstable and tends to agglomerate into separated islands of silver after high temperature annealing. The introduction of Cu into Ag (Cu/Ag \approx 4:7 or 3:5) improves the device thermal stability and allows the device to undergo high temperature (400 °C) treatment with a small V_{th} variation. Then, with the increase of Cu content (e.g. CuAg(8:3) and pure Cu), the devices exhibit increased V_{th} and ON state resistance after 400 °C annealing, possibly due to the interfacial redox reaction. Also, it is worth noting that with higher Cu content, the ON state resistance of the selector becomes higher and suffers a dramatic degradation after 400 °C annealing (Fig. R10).

Thus, from these perspectives, the recommended Cu/Ag ratio for the electrode is from 3:5 to 4:7. For further details regarding this topic, please refer to the response to the 1st question of reviewer #2.

Fig. R10. I - V characteristics of three compositions of CuAg alloy-based selectors before and after

annealing.

Line 159: Reporting the threshold voltages to three significant figures implies strong repeatability and consistency across all tested devices. Looking at the supplemental materials, these thresholds are very stochastic. Please include error ranges (+/- values) with these stochastic quantities. This should be done for all stochastic quantities or quantities which are averaged over many devices.

Response: We thank the reviewer for the valuable comment. The device threshold voltage we described in the previous submission was the mean value of the statistical results, which is now modified to the mean value plus the standard deviation (Fig. R11 and Fig. S8d in the revised supplementary information) following the reviewer's suggestion.

Fig. R11 (Fig. S8d in the revised supplementary information). The variation of the V_{th} as a function of SiO₂ interlayer thickness.

Lines 165-167: Comment: The level of modelling shown here and in the Supplemental materials is an unexpected bonus in an application paper such as this. Even if this paper were to be reframed as a study of the devices themselves, this analysis would be a very nice addition. This is really nice work.

Response: We thank the reviewer for the kind comment and approval.

Side question: Is there any explanation as to why the Cu-only electrodes have low thresholds before annealing, but only require electro-forming after annealing? Given the amount of detail invested in the DFT modelling of the Ag migration, this would be something interesting to include if there is any

better understanding of the change taking place in the Cu devices. Are there perhaps some new parasitic interface layers being formed in this case?

Response: We thank the reviewer for the valuable comment. In the original manuscript, we carried out the annealing experiments for Cu/Al₂O₃/Cu selectors, which shows the forming voltage is very high after annealing (>20 V). In the new submission, we have also carried out further annealing experiments for Cu/SiO₂/Cu selectors. After annealing, the selectors sometimes exhibit overly strong retention (Fig. R9a–c), sometimes still show normal threshold switching behaviors (Fig. R9d). In all cases, the threshold voltage becomes higher (above 2 V) after annealing.

We speculate that this is related to the reaction of Cu with oxide to produce CuO_x. The interfacial CuO_x layer would not only increase the series resistance and threshold voltage, but also weaken the capillary effect of the electrode upon the metallic filament. Hence, the fast migration of metal ions in and out of the dielectric layer is suppressed by this interfacial layer, leading to the observed changes (threshold voltage increase, stronger retention and even conversion to non-volatile switching).

Lines 174-176: No mention is made of the importance of endurance or retention, which implies that the quantities listed are all that matters.

Response: We thank the reviewer for the valuable comment. We added the results and description of the selector's endurance in Fig. R2 (also see Fig. S9 and corresponding discussion in revised Manuscript page 8, paragraph 1 & 2). The CuAg/SiO₂/CuAg selector shows good endurance under AC and DC voltage stimulations, capable of maintaining threshold switching after >10¹⁰ AC pulses or more than 120 seconds of DC stimulation.

As for retention, the CuAg based selectors are volatile switch and exhibit no retention, which means it will immediately turn off after the applied voltage is removed. The transient behaviors of the selectors turn off process have been added (Fig. R2 and Supplementary Fig. S9).

Lines 224-245: Some phenomenological explanation would be helpful in understanding the VMM case. For example, is there some “back-of-the-envelope” scaling relationship which relates the 1) cell-resistance, 2) the stray resistances, and 3) the array size to predict the magnitude of the errors here? Such a supplement (or a citation to one) would be nice to have in helping the audience understand these simulations. This may be very complicated, but any intuitive explanation you can provide would

be helpful.

Response: We thank the reviewer for the valuable comment. The effects of sneak-path currents and IR drop on cross-point array operations have attracted wide research interests and one major research methodology is to carry out array-level circuit simulations (e.g. A. Chen, *IEEE Trans. Electron Devices* 60, 1318(2013); S. Kim et al., *IEEE Trans. Electron Devices* 61, 2820(2014); A. Chen, *J. Comput. Electron.* 16, 1186(2017); J. Woo et al., *IEEE Trans. VLSI Syst.* 27, 2205(2019); N. Lepri et al., *IEEE Trans. Electron Devices* 69, 1575(2022)).

For example, A. Chen (*IEEE Trans. Electron Devices* 60, 1318(2013); *J. Comput. Electron.* 16, 1186(2017)) established a simulation framework based on matrix algebra to calculate the voltages and currents at each node in a crossbar array. This framework is then applied to study the effects of stray resistances and non-linear devices (e.g., selectors) on the read/write margin of the array. The major conclusions can be summarized as: (1) the sensing margin of the crossbar array is affected by both the array size and the ON/OFF ratio (bigger array size and smaller ON/OFF ratio result in worse sensing margin); (2) IR drop is dominated by the ratio between the wireline/stray resistance and the ON resistance of the memory cell (R_L/R_{on}), larger ratio (10^{-3} or above) will lead to more significant IR drop, worse sensing margin and smaller feasible array size; (3) the insertion of selector devices (1S1R) can significantly improve the feasible array size by suppressing the sneak-path currents and effectively increasing the ON/OFF ratio, leading to improved sensing/write margins (better accessibility for each memory cell in the array); (4) for selector design, the bigger the selection ratio the better (in terms of the feasible array size); (5) the design of crossbar arrays with selector devices requires a comprehensive and quantitative approach that incorporates device characteristics, array parameters, operation conditions, and application specifications.

For neural network applications in particular, J. Woo et al. (*IEEE Trans. VLSI Syst.* 27, 2205(2019)) used a similar approach (Verilog + SPICE) to simulate the crossbar array operations and analyzed the accuracy loss. From the simulations, they found that the IR drop caused by either large stray/line resistance (R_{wire}) or high currents associated with large array size degraded the read-out current, resulting in the inaccuracy of the weighted sum on each bit line (which is the output for this application). Moreover, the use of threshold-switch devices (selectors) allowed more linear $I-V$ response of the output which helped to minimize the inaccuracy.

In this work, we follow similar philosophy to carry out our own simulations of the 1S1R array

operations. Our simulation results (Fig. 3f–h in manuscript and Supplementary Fig. S1) in general are consistent with the above findings from prior arts. With extremely high selection ratio and relatively low R_{on} , we think the selector presented in this paper is beneficial for the proposed applications. Citations to the prior arts have been added to the revised manuscript (Manuscript Ref. 42 & 58).

Other questions on these VMM sims:

- What do the correlation diagrams look like in the smaller 32×32 case? One can see in the supplemental figure that the sneak-path currents are not as critical, but how large are they really?

Response: We thank the reviewer for the valuable comment. The output feature map of the 32×32 case obtained by VMM simulation is shown in Fig. R12 (Supplementary information Fig. S14). Also, the output vectors in the three cases (theoretical output, without selector, and with selector) are plotted together for both 32×32 and 64×64 arrays as shown below. Compared to the 64×64 array, the impact of sneak path currents on output accuracy in the 32×32 array is indeed less critical. Still, incorporation of selectors is always beneficial to the accuracy for such VMM operations.

Fig. R12. a, Output feature map obtained by VMM simulation using one fully connected (FC) layer with 32×32 weight matrix for the theoretical output (left, normalized to the average output current

value), without selector (middle), and with selector (right), respectively. **b**, Output currents of 64×64 arrays corresponding to the feature maps. **c**, Output currents of 32×32 arrays corresponding to the feature maps.

- Are the stimulus voltages applied to be inputs assumed to be unipolar?

Response: In these simulations, positive voltages are applied to the selected input lines and 0 V is applied to the selected output line, while unselected lines are set as floating. So yes, the inputs are indeed unipolar.

Line 307: A quick document search indicates that this is the only instance of the word “endurance” in the entire body of the text. The authors claim without evidence that the selector endurance is acceptable, without any reporting of the device endurance, and no arguments or citations as to what level of endurance is required for these applications.

Response: We thank the reviewer for the valuable comment. We added the results and description of the selector’s endurance in Fig. R2 (also see Fig. S9 and corresponding discussion in revised Manuscript page 8, paragraph 1). The CuAg/SiO₂/CuAg selector shows good endurance under AC and DC voltage stimulations, capable of maintaining threshold switching after $>10^{10}$ AC pulses or more than 120 seconds of DC stimulation.

REVIEWER COMMENTS

Reviewer #1 (Remarks to the Author):

The revised manuscript has been improved to large extent and the authors have addressed most questions raised in the previous round of review. In particular, I appreciate the authors' honesty in pointing out the low yield of the 64*64 array and showing the experimental results from a 4*4 sub-array. However, this sub-array is only 1/256 of the whole array, and the 64*64 array in Fig. 3 was fabricated in experiment but not tested in full. This can still be misleading. I urge the authors to show some statistical results on the 64*64 array, e.g. sort of bitmap results, to illustrate the actual yield. Even if the yield is not 100%, the readers would find it helpful for understanding the real performance of the array.

Reviewer #2 (Remarks to the Author):

In the response letter, the authors respond to the questions and suggestions detailedly. Some questions have been worked out. The authors have investigated the atomic ratio of alloy, the switching speed, and the endurance of the devices. However, the problems of experimental measurement and application demonstration of the 64×64 1S1R array have not been well solved yet. The authors should provide additional experimental data to answer the questions before the publication in Nature Communications. The followings are details.

1. For the yield and the experimental performance comparison of the 64×64 1S1R array, the authors said that it is difficult to achieve a high yield of large-scale arrays subject to the technological capability of a university fab. However, the authors seem to have misunderstood the meaning. The opinion in the review is that the comparisons in Fig.3h are expected in the fabricated 64×64 1R arrays experimentally. If it is due to technical reasons, the author could also give a comparison of experimental results on a smaller scale to prove the correctness of the simulation qualitatively.

2. For neuromorphic computing applications, the authors only conducted some single devices tests as well as the physical analog of the LIF model. For both VMM and SNN applications, a large number of devices are required to work together to achieve initial functionality. However, the author's experimental results do not show any indication that these device arrays work together for either VMM or SNN. The authors should use device arrays for more novel SNN-related applications, which is also a good discussion for the endurance and thermal stability required in the array applications, instead of the validation of the selector-based LIF neuron.

Reviewer #3 (Remarks to the Author):

This reviewer would like to thank the authors for the careful responses to the points raised in my original review. The revised version is greatly improved. I have no other critiques.

Reviewer #1:

The revised manuscript has been improved to large extent and the authors have addressed most questions raised in the previous round of review. In particular, I appreciate the authors' honesty in pointing out the low yield of the 64*64 array and showing the experimental results from a 4*4 sub-array. However, this sub-array is only 1/256 of the whole array, and the 64*64 array in Fig. 3 was fabricated in experiment but not tested in full. This can still be misleading. I urge the authors to show some statistical results on the 64*64 array, e.g. sort of bitmap results, to illustrate the actual yield. Even if the yield is not 100%, the readers would find it helpful for understanding the real performance of the array.

Response: We thank the reviewer for the valuable comment. In order to better illustrate the actual yield situation, we carried out further testing on a total of 144 1S1R devices in the same 64×64 array, as shown in Fig. R1 (also Fig. S14 in the revised supplementary information). In the subarray tested, 52 out of the 144 devices (green counts) exhibited the complete 6-characteristic transition processes of 1S1R, 21 devices (purple counts) showed selector-only characteristic due to shorted memristor, 44 devices (orange counts) exhibited memristor-only characteristic due to shorted selector, and 27 devices (red counts) showed open-circuit state with no switching at high operating voltage.

It should be noted that for those open devices in the 64×64 array, the leakage current is as low as $10^{-11} \sim 10^{-12}$ A (instrument limit), indicating that the selectors have successfully suppressed the sneak-path leakage currents in a very large array. The overall yield of 1S1R devices in the 12×12 subarray is 36%, and the yield bitmap is shown in Fig. R1 (also Fig. S14 in the revised supplementary information). We added relevant descriptions in page 10, paragraph 1 of the manuscript, associated with the representative data in the revised supplementary information (Fig. S14).

Fig. R1 (Fig. S14 in the revised supplementary information). Exploration of the yield of the 1S1R array. a1, a2, The electrical properties of 144 devices in the 64×64 1S1R array. Green counts: 1S1R devices with complete 6 switching processes; purple counts: devices with only normal operation of the selector and shorted memristors; orange counts: devices with only normal operation of the memristors and shorted selectors; red counts: devices with higher operating voltage and presenting open circuits. b, The bit map corresponding to 144 devices.

Reviewer #2:

In the response letter, the authors respond to the questions and suggestions detailly. Some questions have been worked out. The authors have investigated the atomic ratio of alloy, the switching speed, and the endurance of the devices. However, the problems of experimental measurement and application demonstration of the 64×64 1S1R array have not been well solved yet. The authors should provide additional experimental data to answer the questions before the publishment in Nature Communications. The followings are details.

1. For the yield and the experimental performance comparison of the 64×64 1S1R array, the authors

said that it is difficult to achieve a high yield of large-scale arrays subject to the technological capability of a university fab. However, the authors seem to have misunderstood the meaning. The opinion in the review is that the comparisons in Fig.3h are expected in the fabricated 64×64 1R arrays experimentally. If it is due to technical reasons, the author could also give a comparison of experimental results on a smaller scale to prove the correctness of the simulation qualitatively.

Response: We thank the reviewer for the valuable comment. We have carried out further experimental characterizations of the 1R array and compare it with the simulations. First of all, as shown in Fig. R2a, we characterized the DC I - V characteristics of a single 1R device and an 8×8 1R subarray in a 64×64 1R array without selectors. During 1R array measurements, the voltages were applied between the selected input and output lines, while all unselected input and output lines were floated. It can be observed that the effective ON/OFF ratios are severely degraded in the 64×64 1R array due to the presence of sneak-path currents.

We then use this 8×8 1R subarray to carry out VMM operations and compare with the 1S1R array. The detailed methods are as follows: a probe card is designed to simultaneously probe the 8 input and 8 output lines of the 1R subarray (Fig. R2b). The 16 input/output channels are connected with the 16 output terminals of a Keithley 707A switch matrix containing two Keithley 7174A low leakage matrix cards (each with 8 input and 12 output channels). Three channels of the Keithley 707A switch matrix cards are connected to three SMU modules of a Keithley 4200 SCS. Among them, two SMU channels connected to the input lines through the switch matrix, and generates high and low input voltages, respectively. This way, we can generate arbitrary binary input vectors for the 8 input lines. The other SMU is connected to the output lines through the switch matrix, and can be used to measure the output current while forcing 0 V voltage on a selected output line.

Then, we randomly generated 8 sets of 8×8 binary weight matrices and programmed them into the 8×8 array one by one. For each weight matrix, all $2^8=256$ possible inputs are applied using the switch matrix and the output current vectors for each input are measured. In order to evaluate the output accuracy, we concatenate the output vectors of each weight matrix to make a 64-long output vector for each input. The VMM accuracy can be evaluated by calculating the correlation coefficients of each output vector versus the theoretical output:

$$CorrCoef = \frac{cov(I_{measure,k}, I_{theory})}{\sqrt{var(I_{measure,k}) \times var(I_{theory})}}$$

in which, $I_{\text{measure},k}$ ($k=1,\dots,255$) is the measured output current vector for each input k , I_{theory} is the theoretical output calculated as shown in Fig. R2c. Fig. R2d (left) shows the results of the correlation coefficient with 8×8 1R array, which is averaged to 84.15%.

Finally, we carried out similar experiments with the 1S1R array. However, due to the yield issue of the 1S1R array, we were only able to find working 4×4 1S1R array. Thus, we break each 8×8 weight matrix into four 4×4 weight matrices and carry out the output vector measurements four times. After that, we concatenate the outputs to obtain the same output format as 1R array. In this case, the correlation coefficient is improved to an average of 93.93%, as shown in Fig. R2d.

Also, the probability density of the correlation coefficients of the 8×8 array obtained by the same simulation method as the Fig. 3h is shown in Fig. R2e. It is worth noting that during the theoretical simulations, the HRS and LRS resistances in 1R and 1S1R arrays are set to be constant values (average value of resistance in HRS/LRS), thus the accuracy obtained from the calculation is higher than that of the experiment. From the experimental results we can see that the HRS and LRS resistance variations can also contribute to degradation of accuracies, especially in the case of 64×64 1R array. Based on these results, we may conclude that 1S1R allows for a high accuracy way of performing VMM operation. We added relevant descriptions in page 11, paragraph 2 of the manuscript, associated with the representative data in the revised supplementary information (Fig. S17).

Fig. R2 (Fig. S17 in the revised supplementary information). Accuracy verification of the subarray in the 64×64 array for VMM calculations. a, The operating window of 1R devices tends to decrease as the array size increases. b, Schematic diagram of VMM operation test structure. c, Schematic diagram of the cross-point array performing VMM. d, Distribution of correlation coefficients (left) between the output results obtained using 8-bit binary from 1 to 255 as input and the

theoretical output results, and the distribution of 64 output currents (right) obtained after using 8 weight settings for one of the 255 groups. e, The probability density of the correlation coefficients obtained by the same simulation method as the Fig. 3h.

2. For neuromorphic computing applications, the authors only conducted some single devices tests as well as the physical analog of the LIF model. For both VMM and SNN applications, a large number of devices are required to work together to achieve initial functionality. However, the author's experimental results do not show any indication that these device arrays work together for either VMM or SNN. The authors should use device arrays for more novel SNN-related applications, which is also a good discussion for the endurance and thermal stability required in the array applications, instead of the validation of the selector-based LIF neuron.

Response: We thank the reviewer for the valuable comment. It is indeed a very good and insightful advice that the proposed 1S1R array could be more suitable for SNN-related applications. In the answer to the above question, we have already demonstrated that for VMM applications, 1S1R array can generate more accurate outputs compared to 1R-only array especially at a large array size. In order to explore its potential applications in SNN, we further studied the pulse response of 1S1R and 1R synaptic arrays experimentally. As shown in Fig. R3 (also Fig. S15 in the revised supplementary information), we mimic the input pulses of a SNN with a Keysight 33250A waveform generator and send the signal to one of the input lines (word line) in the 64×64 1S1R or 1R arrays, while all other input lines are floated. On the output side, we use a voltage-dividing resistor R_0 (10 k Ω , approximately equals to the LRS resistance) to connect the selected output line (bit line) and the ground for simplicity. An Agilent MSO7054A oscilloscope is used to capture the voltage waveforms of the input line, the selected and unselected output lines to gain a comprehensive understanding.

From the results, we had some interesting discoveries:

- (1) For 64×64 1S1R array, the multiplication result of the input voltage versus the weight in terms of RRAM conductance is clearly reflected in the output. Compared to the HRS case, LRS can pass higher current/voltage to the output. Moreover, the unwanted disturbance to unselected output lines is negligible thanks to the presence of selectors which suppress sneak-path currents. The output signal's delay time with reference to the input signal (at 50% voltage range) is 0.02 and 0.73 μs for HRS and LRS, respectively.

(2) For 64×64 1R array, the signal on the output line exhibits a longer rise and fall time. The HRS and LRS output delay time is 1.48 and $3.44 \mu\text{s}$, respectively. This suggests stronger parasitic effects during the propagation of pulse signal. Moreover, the voltage on the unselected output line also exhibits a slow but significant increase in both HRS and LRS cases, suggesting that the voltage of unselected lines is strongly pulled by the input line due to the lack of selectors and thus the abundance of sneak-path currents.

These observations indicate that 1S1R array is a particularly useful technology for SNN applications, which not only improves the output accuracy, but also shields the sneak-path currents and parasitic capacitance on unselected lines to significantly improve the operation speed and reduce the power consumption. The corresponding discussion is added in page 10, paragraph 1 of the manuscript, associated with the representative data in the revised supplementary information (Fig. S15).

Fig. R3 (Fig. S15 in the revised supplementary information). Verification circuits of SNN operations (forward propagation of pulses) with 1S1R/1R array as synapses. a, Schematic diagram of the verification circuit using a pulse of amplitude 1 V input to the word line of the selected

device when the selected device is in HRS and LRS, respectively, and simultaneously reading the selected device bit line and nearest bit line with an oscilloscope. **b**, Schematic diagram of read operation for selected devices in 64×64 1S1R and 1R arrays. **c**, Output results of selected bit line and nearest unselected bit line in 64×64 1S1R array and 1R array under a pulse input of 1 V amplitude.

Reviewer #3 (Remarks to the Author):

This reviewer would like to thank the authors for the careful responses to the points raised in my original review. The revised version is greatly improved. I have no other critiques.

Response: We thank the reviewer for taking time to review this work and for the positive comments to this work.

REVIEWERS' COMMENTS

Reviewer #1 (Remarks to the Author):

The authors have conducted additional work to address the questions with detailed experimental data. Although the device yield is not high, I found the overall responses acceptable. I have no further questions and can recommend publication.

Reviewer #2 (Remarks to the Author):

The authors respond to the questions raised in the last review carefully in the latest revised version. The manuscript has been greatly improved. I have no further comments.

Reviewer #1:

The authors have conducted additional work to address the questions with detailed experimental data. Although the device yield is not high, I found the overall responses acceptable. I have no further questions and can recommend publication.

Response: We thank the reviewer for taking time to review this work and for the positive comments to this work. We will continue to delve into the manufacture and application of 1S1R to enhance our understanding of the novel memory.

Reviewer #2:

The authors respond to the questions raised in the last review carefully in the latest revised version. The manuscript has been greatly improved. I have no further comments.

Response: We thank the reviewer for taking time to review this work and for the positive comments to this work. We look forward to unveiling more application possibilities of 1S1R in neuromorphic computing.